# TNFSF14$^+$ natural killer cells prevent spontaneous abortion by restricting leucine-mediated decidual stromal cell senescence

Jia-Wei Shi[1,2,3], Zhen-Zhen Lai[2], Wen-Jie Zhou[4], Hui-Li Yang[2], Tao Zhang [ID][5], Jian-Song Sun[6], Jian-Yuan Zhao [ID][7✉] & Ming-Qing Li [ID][1,2,8✉]

## Abstract

**In preparation for a potential pregnancy, the endometrium of the uterus changes into a temporary structure called the decidua. Senescent decidual stromal cells (DSCs) are enriched in the decidua during decidualization, but the underlying mechanisms of this process remain unclear. Here, we performed single-cell RNA transcriptomics on ESCs and DSCs and found that cell senescence during decidualization is accompanied by increased levels of the branched-chain amino acid (BCAA) transporter SLC3A2. Depletion of leucine, one of the branched-chain amino acids, from cultured media decreased senescence, while high leucine diet resulted in increased senescence and high rates of embryo loss in mice. BCAAs induced senescence in DSCs via the p38 MAPK pathway. In contrast, TNFSF14+ decidual natural killer (dNK) cells were found to inhibit DSC senescence by interacting with its ligand TNFRSF14. As in mice fed high-leucine diets, both mice with NK cell depletion and Tnfrsf14-deficient mice with excessive uterine senescence experienced adverse pregnancy outcomes. Further, we found excessive uterine senescence, SLC3A2-mediated BCAA intake, and insufficient TNFRSF14 expression in the decidua of patients with recurrent spontaneous abortion. In summary, this study suggests that dNK cells maintain senescence homeostasis of DSCs via TNFSF14/TNFRSF14, providing a potential therapeutic strategy to prevent DSC senescence-associated spontaneous abortion.**

**Keywords** TNFRSF14; Aging; Decidualization; Leucine; Abortion
**Subject Categories** Development; Immunology; Stem Cells & Regenerative Medicine

## Introduction

Decidualization is the differentiation and transformation of the endometrial stroma, which is crucial for embryo implantation, placental development, and maternal-fetal interface establishment (Gellersen and Brosens, 2014). Defective or impaired decidualization has several clinical consequences, including implantation failure, spontaneous abortion in early pregnancy, preeclampsia during advanced gestation, and fetal growth restriction (FGR) (Brosens et al, 2019; Garrido-Gomez et al, 2022; Gellersen and Brosens, 2014; Shi et al, 2022; Staff et al, 2022). Despite significant progress in assisted reproductive technology, there is still no effective prevention for pregnancy loss and especially recurrent implantation failure (Gellersen and Brosens, 2014). Therefore, performing an in-depth exploration of decidualization will contribute to the development of novel therapeutic targets and strategies.

Cell senescence refers to the gradual decline in cell proliferation and differentiation among aged or damaged cells, ultimately leading to permanent cell cycle arrest and the loss of normal physiological functions (Hernandez-Segura et al, 2018). Senescent cells exhibit morphological (an enlarged hypertrophic morphology) and molecular features (such as expression of senescence markers, CDKN2A, CDKN1A, TP53), along with the enhanced lysosomal hydrolase activity, termed senescence-associated β-galactosidase (SAβG) (Hernandez-Segura et al, 2018). Metformin, resveratrol, and rapamycin are reported as anti-aging drug candidates that delay or improve the aging process (Chen et al, 2022; Grinan-Ferre et al, 2021; Kulkarni et al, 2020; Yousefzadeh et al, 2021). Recent evidence has indicated that decidualization is accompanied by the appearance of stromal cells with senescent-like characteristics (Deryabin et al, 2020; Hou et al, 2023; Lucas et al, 2016; Lucas et al, 2020; Shi et al, 2022; Zeng et al, 2023). However, the regulatory mechanisms underlying stromal senescence during decidualization and its role in spontaneous abortion remain largely unknown.

[1]Department of Reproductive Immunology, The International Peace Maternity and Child Health Hospital, School of Medicine, Shanghai Jiao Tong University, Shanghai 200030, People's Republic of China. [2]Laboratory for Reproductive Immunology, Hospital of Obstetrics and Gynecology, Shanghai Medical School, Fudan University, Shanghai 200080, People's Republic of China. [3]Department of Obstetrics and Gynecology, The first affiliated Hospital of Ningbo University, Ningbo 315021, People's Republic of China. [4]Reproductive Medical Center, Department of Obstetrics and Gynecology, Ruijin Hospital, Shanghai Jiao Tong University School of Medicine, Shanghai 200025, People's Republic of China. [5]Assisted Reproductive Technology Unit, Department of Obstetrics and Gynecology, Faculty of Medicine, Chinese University of Hong Kong, Hong Kong, People's Republic of China. [6]School of Life Science and Health Engineering, Jiangnan University, Wuxi 214122, People's Republic of China. [7]Institute for Developmental and Regenerative Cardiovascular Medicine, MOE-Shanghai Key Laboratory of Children's Environmental Health, Xinhua Hospital, Shanghai Jiao Tong University School of Medicine, Shanghai 200092, People's Republic of China. [8]Shanghai Key Laboratory of Embryo Original Diseases, Shanghai 200030, People's Republic of China.
✉E-mail: zhaojianyuan1508@xinhuamed.com.cn; mqli@sjtu.edu.cn

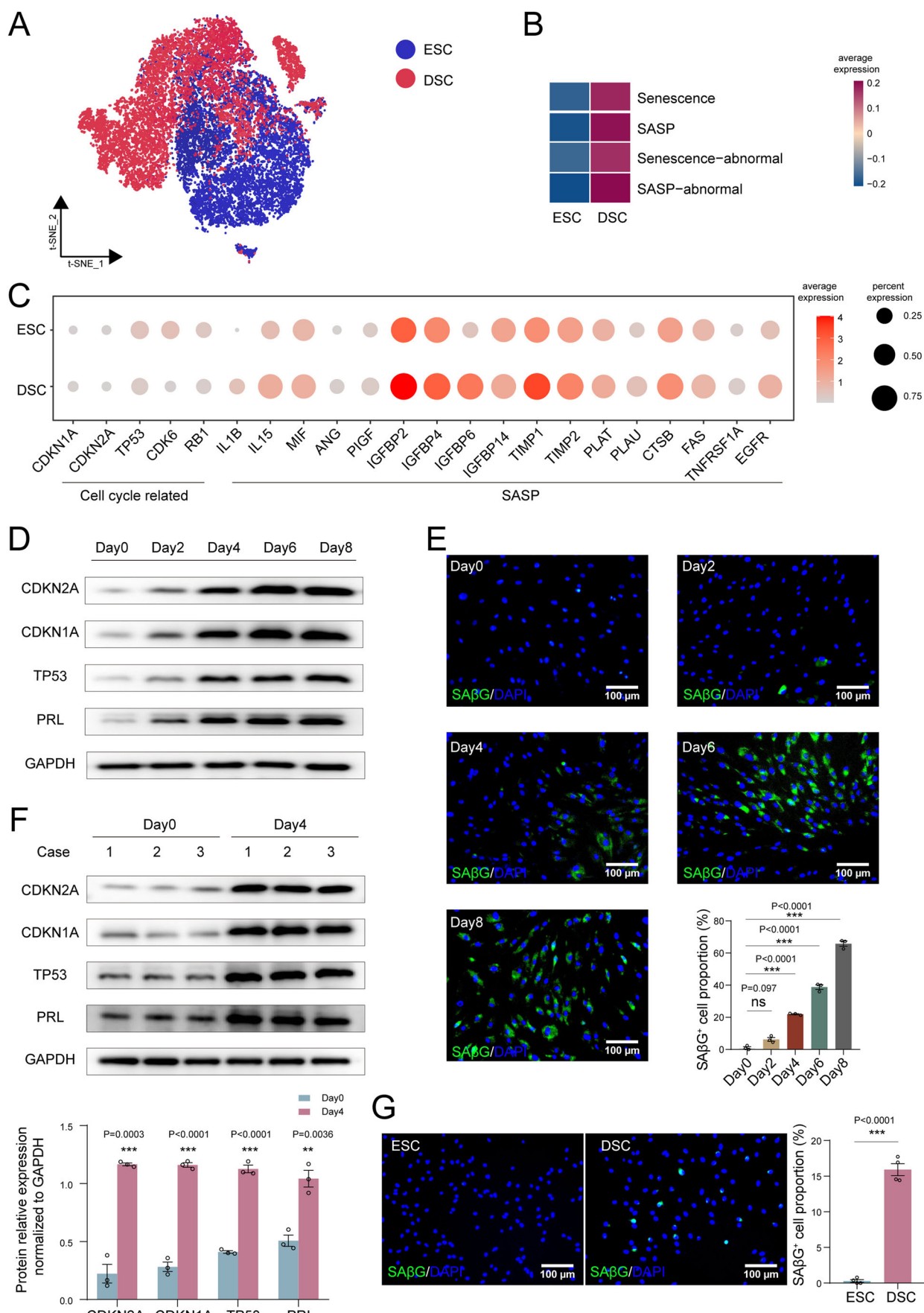

**Figure 1.   The decidualization process is accompanied with senescent DSCs.**

(A) Subpopulation of stromal cells of secretory endometrium and decidua in t-SNE plots through single-cell sequencing. (B) QuSAGE analysis of ESCs and DSCs through single-cell sequencing. (C) Bubble diagram presented the average expression of cell cycle and senescence-associated secretory phenotype (SASP) related genes based on the results of single-cell sequencing. (D) Human ESC cell line (hESCs) was treated with 8-bromo-cAMP (0.5 mM) plus MPA (1 μM) for different times, and western blotting assay was used to detect the expression of CDKN2A, CDKN1A, TP53 (indicator of cell senescence) and PRL. (E) SAβG staining (indicator of cell senescence) and statistical data of decidualized hESCs ($n = 3$ biological replicates per group). (F) Primary ESCs were treated with 8-bromo-cAMP (0.5 mM) plus MPA (1 μM) for 4 days, and the protein of CDKN2A, CDKN1A, TP53, and PRL were measured by western blotting ($n = 3$ biological replicates per group), relative expression levels of proteins were standardized using internal reference GAPDH. (G) SAβG staining in primary ESCs ($n = 4$ biological replicates) and primary DSCs ($n = 4$ biological replicates). Statistical data were presented as mean ± SEM. $^{**}P < 0.01$, $^{***}P < 0.001$, ns, no significance, using one-way ANOVA with Bonferroni multiple comparisons test (E), using a two-tailed, unpaired Student's t test (F, G). Source data are available online for this figure.

Decidualization is related to a range of genetic, metabolic, morphological, biochemical, vascular, and immune changes in the endometrial stroma (Shi et al, 2022; Tang et al, 2023; Wang et al, 2021). Endometrial stromal cells (ESCs) differentiate into decidual stromal cells (DSCs) in the presence of estrogen and progesterone. The maternal-fetal interface is mainly composed of trophoblasts, DSCs, and decidual immune cells (DICs) (Gellersen and Brosens, 2014). Human NK cells constitute approximately 50–70% of the lymphocytes in the decidua (Vacca et al, 2011); these decidual NK cells (dNK) are involved in immune regulation, decidualization, embryo implantation, fetal growth, placental development, and angiogenesis (Crespo et al, 2020; Male and Moffett, 2023; Vacca et al, 2011). Accumulating evidence indicates that NK cells facilitate the clearance of senescent cells and maintain tissue homeostasis (Deryabin et al, 2020; Jin et al, 2021; Pereira et al, 2019). Notably, increased autophagy and stromal cell adhesion during decidualization lead to extensive NK cell infiltration and enrichment, particularly of TNF superfamily member 14-positive (TNFSF14[+], also named LIGHT[+]) NK cells (Lu et al, 2021). TNFSF14, also known as a T-cell costimulatory molecule, regulates fibroblastic reticular cell senescence by binding to its ligand, TNFRSF14 (also known as HVEM) (Li et al, 2020). However, the roles of dNK cells in the senescence homeostasis of stromal cells at the maternal-fetal interface remain unclear.

More recently, metabolic reprogramming, such as that of nicotinamide adenine dinucleotide (NAD[+]) and branched chain amino acid (BCAA: leucine, valine, and isoleucine) metabolism, is considered directly and indirectly involved in cell senescence (Igelmann et al, 2021; Le Couteur et al, 2020; Miwa et al, 2022; Moller et al, 2022). Therefore, we speculate that BCAAs may be involved in DSC senescence during decidualization, whereas NK cells can maintain the homeostasis of DSC senescence at the maternal-fetal interface. The present study aimed to investigate the immunometabolic mechanisms of DSC senescence during early pregnancy and its pathogenic role in recurrent spontaneous abortion (RSA) to identify potential intervention strategies both in vitro and in vivo.

# Results

## Excessive senescence of DSCs leads to pregnancy loss

Based on our single-cell sequencing data (Shi et al, 2022), levels of cell senescence markers between total stromal cells in the endometrium and decidua, including cell cycle-related (such as CDKN2A, CDKN1A, and TP53) and senescence-associated

secretory phenotype (SASP)-related genes (that is, IL1B, IL15, IGFBP2, TIMP1), were analyzed. Increased cell senescence levels were observed between ESCs and DSCs (Fig. 1A–C). To evaluate stromal cell senescence levels during decidualization, we constructed in vitro decidualization models of human ESC (hESC) cell line and primary ESCs. Following exposure to 8-bromo-cAMP and medroxyprogesterone acetate (MPA), cell senescence (marker genes: CDKN2A, CDKN1A, and TP53), and decidualization (marker gene: PRL) levels in hESCs increased significantly in a time-dependent manner, similar to SAβG activity (a common biomarker for senescent cells) (Fig. 1D–F) (Debacq-Chainiaux et al, 2009). Prostaglandin $E_2$ (PGE$_2$) is another decidualization inducer (Stadtmauer and Wagner, 2022); the PGE$_2$-induced decidualization is also accompanied by increased expression of senescence-related molecules in hESC cells (Fig. S1A). Similarly, primary DSC displayed an increased level of senescence compared with primary ESC (Fig. 1G).

Interestingly, we observed increased levels of CDKN2A, CDKN1A, and TP53 in DSCs from females with unexplained RSA using both immunohistochemistry (Fig. 2A) and western blotting (Fig. 2B). Together with the fact that SAβG activity increased in DSCs from patients with RSA (Fig. 2C), these results suggested excessive senescence of DSCs from patients with RSA. To further study the potential role of excessive senescence in abortion, an aging mouse model was constructed by hypodermic injections of D-galactose over 8 weeks (Fig. S1B,C). Blastocysts obtained from pregnant C57BL/6 mice of normal reproductive age were transplanted into the uteri of control mice and D-galactose-treated mice, and seven blastocysts were transplanted into each side of their uteri. Notably, pregnant mice with excessive uterine senescence were prone to embryo loss (increased absorption rate) along with fewer blastocyst implantations (Fig. 2D–G). These data indicate that decidualization is accompanied by the appearance of senescent DSC subsets, and the abnormal accumulation of senescent DSCs increases the risk of spontaneous abortion.

## Leucine accumulation induces excessive senescence of DSC and spontaneous abortion

Single-cell sequencing-based differential gene enrichment analysis between ESCs and DSCs was conducted to explore the mechanisms of stromal senescence during decidualization. In addition to DSC senescence, we observed an increase in BCAA transport (i.e., solute carrier family 3 member 2 [SLC3A2]) and oxidative stress (i.e., glutathione peroxidase 1 [GPX1] and reactive oxygen species [ROS] production) during decidualization (Fig. 3A; Fig. S2A). SLC3A2 interacts with SLC7A5 to form a large neutral amino acid (LNAA)

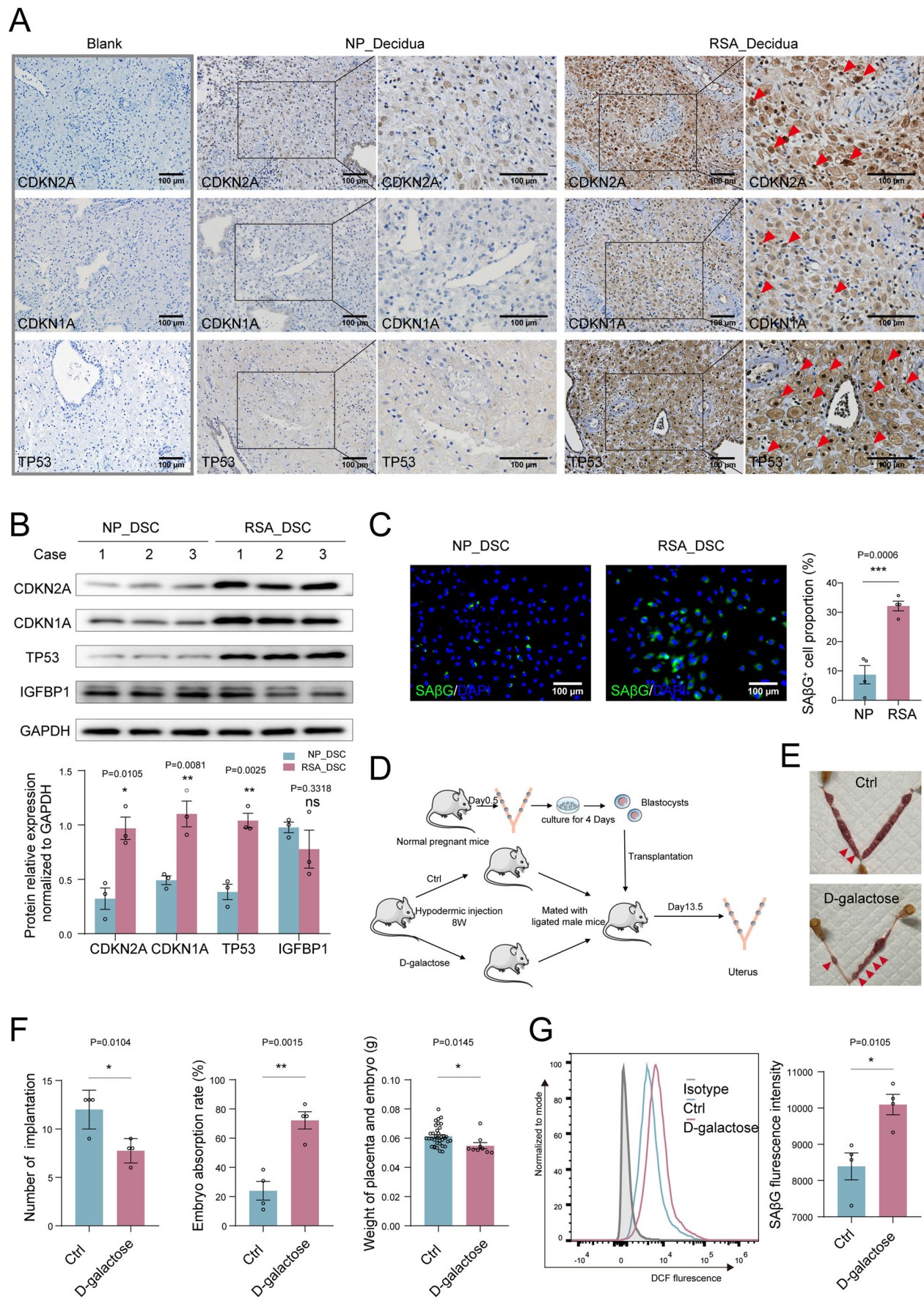

**Figure 2.  Excessive senescence of DSCs leads to pregnancy loss.**

(A) Expression of CDKN2A, CDKN1A, and TP53 in the decidual from normal pregnancy (NP) and females with unexplained recurrent spontaneous abortion (RSA) were analyzed by immunohistochemistry ($n = 6$ biological replicates for each group). Scale bar, 100 μm. (B) The expression of CDKN2A, CDKN1A, TP53, and IGFBP1 in DSCs of NP ($n = 3$ biological replicates) and RSA ($n = 3$ biological replicates) was detected by western blotting, relative expression levels of proteins were standardized using internal reference GAPDH. (C) The Immunofluorescence staining of SAβG in DSCs of NP ($n = 4$ biological replicates) and RSA ($n = 4$ biological replicates). Scale bar, 100 μm. (D) Schematic diagram of aging mouse model construction. (E, F) The pregnancy outcome at the gestation of day 13.5 in the control ($n = 4$ biological replicates) and D-galactose injected group ($n = 4$ biological replicates). Number of blastocyst implantation, embryo resorption rate, and weight of the embryo and placenta in the control ($n = 4$ biological replicates) and D-galactose ($n = 4$ biological replicates) group. (G) SAβG activity of DSCs were measured by flow cytometry in the control ($n = 4$ biological replicates) and D-galactose ($n = 4$ biological replicates) group. Data were presented as mean ± SEM. *$P < 0.05$, **$P < 0.01$, ***$P < 0.001$, ns, no significance, using a two-tailed, unpaired Student's t test. NP normal pregnancy, RSA recurrent spontaneous abortion. Source data are available online for this figure.

transporter (Ikeda et al, 2017). More importantly, deprivation of leucine, valine, isoleucine, or total BCAA in the culture medium, or silencing *SLC3A2* obviously decreased the expression levels of CDKN2A, CDKN1A, and TP53, indicating inhibited senescence of decidualized ESCs (Fig. 3B–D). In addition, leucine deficiency or apocynin application (an inhibitor of NADPH oxidase-related ROS synthesis) significantly suppressed ROS production from decidualized ESCs (Fig. 3E), indicating that BCAA intake induces oxidative stress and senescence of ESC during decidualization.

The p38 MAPK signaling pathway regulates cell senescence (He et al, 2020; Yuan et al, 2023). The p38 MAPK signaling pathway was activated during decidualization (Fig. S2B,C). However, leucine deficiency or silencing *SLC3A2* resulted in inactivation of the p38 MAPK signaling pathway in decidualized ESCs (Fig. S2D,E). These findings indicated that SLC3A2-mediated BCAA intake induces senescence in decidualized ESCs in a p38 MAPK signaling pathway-dependent manner.

Further analysis indicated greater SLC3A2 expression and intake of BCAA by DSCs from patients with RSA compared to DSCs from females with normal pregnancy (Fig. 3F,G).

To explore the potential roles of leucine in DSC senescence and pregnancy outcomes, pregnant mice were fed control or high-leucine (6%) chow. As depicted, high-leucine chow led to increased SAβG activity, and elevated expression of CDKN2A, CDKN1A, and TP53 in mice uterine (Fig. 4A,B). In addition, we observed high rates of embryo loss and low embryo and placental weights in high-leucine-fed pregnant mice (Fig. 4C,D). These findings indicate that abnormally high levels of SLC3A2 and leucine accumulation contribute to excessive senescence in DSC and spontaneous abortion, possibly dependent on the p38 MAPK signaling pathway.

## dNK cells maintain DSC senescence homeostasis and normal pregnancy in a TNFSF14/TNFRSF14-dependent manner

The dNK cells regulate decidual homeostasis (Wang et al, 2021), but its role in DSC senescence remains unclear. Here, we observed that co-culturing DSCs with dNK cells significantly decreased DSC senescence (Fig. 4E; Fig. S3A–C). Our previous studies showed that TNFRSF14 is highly expressed in DSCs and is involved in the residence and enrichment of dNK cells in the first trimester (Lu et al, 2021). Among the ligands of TNFRSF14, including TNFSF14, CD160, and BLTA, TNFSF14 was the most highly expressed in the decidua at the maternal-fetal interface, especially in NK and T cells (Fig. S4A,B). Compared to the decidua of females with normal pregnancy, those of patients with RSA displayed an obvious decrease in TNFRSF14 levels; however, the percentage of

TNFSF14+ dNK cells did not change significantly (Fig. 4F,G). Notably, silencing *TNFRSF14* decreased the inhibition of DSC senescence by dNK cells (Fig. 4H,I; Fig. S4C,D), indicating that dNK cells maintain DSC senescence homeostasis via the TNFSF14/TNFRSF14 axis. More importantly, excessive decidual senescence, increased embryo absorption, and decreased placenta weight were detected in *Tnfrsf14*-knockout pregnant mice (*Tnfrsf14*$^{-/-}$) (Fig. 5A–D) further demonstrating the vital role of TNFRSF14 in maintaining normal pregnancy.

## TNFSF14+ dNK cells control DSC senescence by the BCAA/p38 MAPK signaling pathway

To investigate the underlying mechanisms of TNFSF14+ dNK cells in DSC senescence, we investigated an in vitro decidualization model of hESCs that was treated with human recombinant TNFSF14 protein for 48 h (Fig. 5E). TNFSF14 markedly decreased decidualized stromal cell senescence (Fig. 5F,G). More excitingly, RNA sequencing analysis suggested that differentially expressed genes in the TP53 and MAPK signaling pathways, DNA replication, senescence, and cell cycle were significantly enriched in the TNFSF14-treatment group, as well as a decrease in SLC3A2 and heat shock factor 1 (HSF1) (Fig. 6A,B). Further verification showed that both TNFSF14 and dNK cells decreased SLC3A2 and BCAA intake in decidualized ESCs; however, silencing *TNFRSF14* could suppress these effects (Fig. 6C–G). Moreover, TNFSF14 and dNK cells inhibited activation of the p38 MAPK pathway (Fig. S5A,B). Subsequently, the relationship between HSF1 and SLC3A2 was verified using a double luciferase reporter assay (Fig. S5C; Fig. 6H,I), indicating that LIGHT inhibited SLC3A2 expression by decreasing HSF1 levels. Therefore, these results suggest that TNFSF14 downregulates SLC3A2 expression and BCAA intake levels and further controls DSC senescence by inactivating the p38 MAPK pathway.

## TNFSF14 and metformin reduce the risk of excessive DSC senescence-associated pregnancy loss

To identify the potential value of intervening in DSC senescence as a treatment for spontaneous abortion, we first constructed an NK cell-depleted mouse model using an anti-NK1.1 neutralizing antibody (Fig. S6A,B). Of note, a deficiency of NK cells upregulated CDKN2A, CDKN1A, and TP53 expression as well as SAβG activity in uterine of pregnant mice (Fig. 7A,B). In addition, NK cell deletion induced adverse pregnancy outcomes, including higher rates of pregnancy loss, and placental and embryonic disorders (Fig. 7C,D). As expected, TNFSF14 protein alleviated

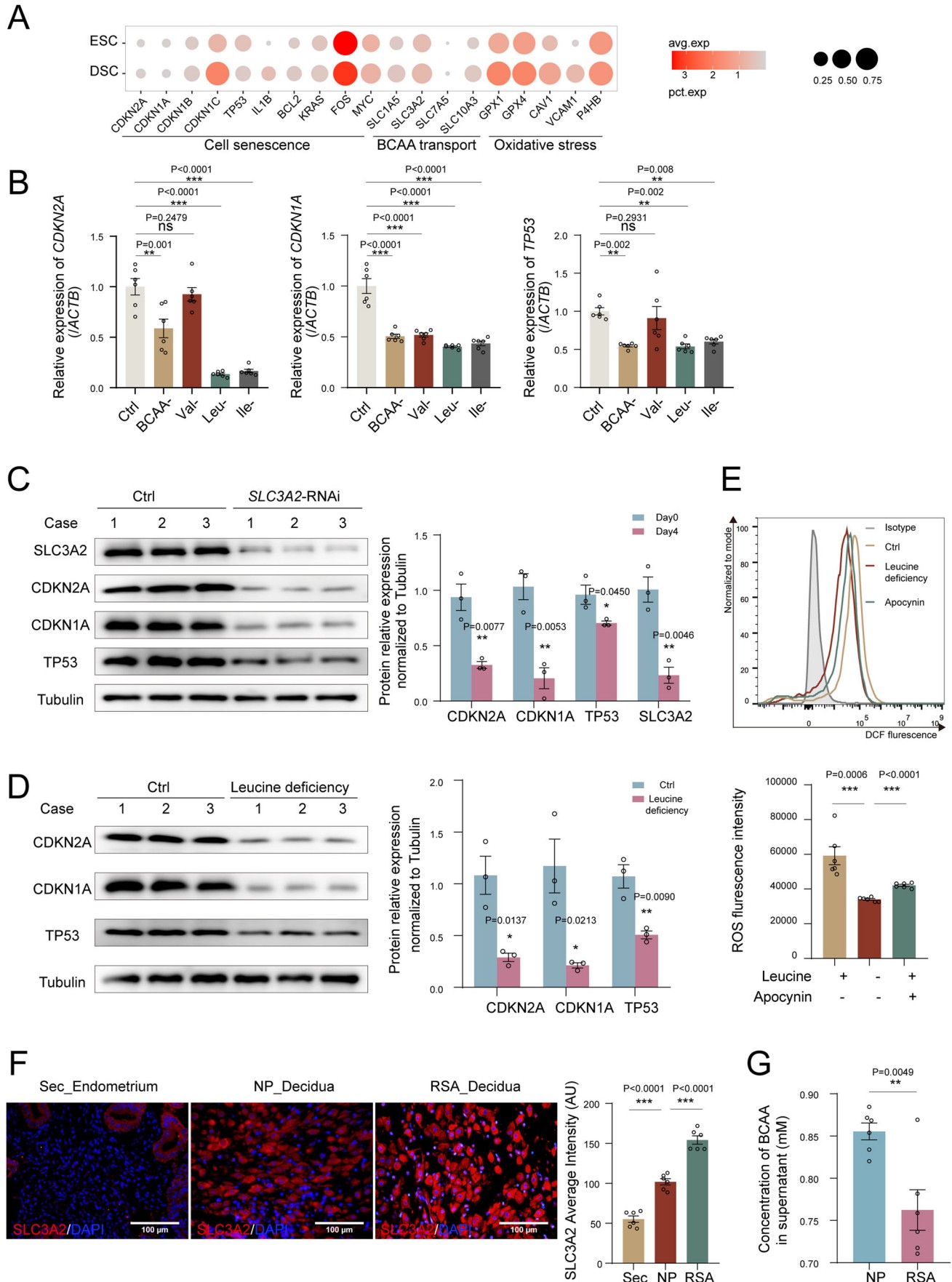

**Figure 3. Leucine accumulation induces excessive senescence of DSC and spontaneous abortion.**

(A) Bubble diagram showing the average expression of BCAA transport and oxidative stress-related genes of stromal cells from control endometrium (ESC) and decidua of normal pregnancy (DSC) through single-cell sequencing. (B) During the process of decidualization (cAMP plus MPA), hESCs were cultured with medium lacking BCAA, leucine, valine, or isoleucine, respectively, for 96 h. The mRNA expression of *CDKN2A*, *CDKN1A*, and *TP53* were detected by qRT-PCR (*n* = 6 biological replicates per group). (C) During decidualization, the expression of SLC3A2, CDKN2A, CDKN1A, and TP53 in control and si*SLC3A2* hESCs were measured by western blotting (*n* = 3 biological replicates per group), relative expression levels of proteins were standardized using internal reference Tubulin. (D) hESCs were cultured with media of leucine deficiency during decidualization, and CDKN2A, CDKN1A, and TP53 expression were detected by western blotting (*n* = 3 biological replicates per group), relative expression levels of proteins were standardized using internal reference Tubulin. (E) hESCs were cultured with normal media, leucine deficiency media, or normal media plus apocynin (10 μM) during decidualization, and ROS were measured by flow cytometry (*n* = 6 biological replicates per group). (F) The expression of SLC3A2 in ESCs of secretory phase endometrium, DSCs of NP and RSA were detected by immunofluorescence (*n* = 6 biological replicates per group). Scale bar, 100 μm. (G) Concentration of BCAA in supernatant of DSCs form NP (*n* = 6 biological replicates) and RSA (*n* = 6 biological replicates) were measured by BCAA detection kit. NP normal pregnancy, RSA recurrent spontaneous abortion. Data were presented as mean ± SEM. *$P < 0.05$, **$P < 0.01$, ***$P < 0.001$, ns, no significance, using one-way ANOVA with Bonferroni multiple comparisons test (B), using a two-tailed, unpaired Student's t test (C–G). Source data are available online for this figure.

uterine senescence and reduced the risk of adverse pregnancy outcomes (Fig. 7A–D).

Considering that no change in TNFSF14 was observed in patients with RSA (Fig. 4B), we next discovered additional downstream regulatory measures of DSC senescence to prevent pregnancy loss. Metformin is considered an anti-aging drug in humans (Chen et al, 2022; Kulkarni et al, 2020). We observed that metformin decreased DSC senescence in patients with RSA in vitro (Fig. 8A). In a D-galactose-induced aging mouse model, treatment with metformin downregulated uterine senescence, including lower levels of CDKN2A, CDKN1A, and TP53, and decreased SAβG activity (Fig. S7A,B). Notably, metformin reduced the risk of adverse pregnancy outcomes in D-galactose-induced senescent mice, including a higher number of blastocyst implantations, greater placenta and embryo weights, and lower rates of pregnancy loss (Fig. 8B,C). Remarkably, the protective effects of metformin against vimentin+ uterine stromal cell senescence and pregnancy loss were also observed in *Tnfrsf14*−/− pregnant mice (Fig. 8D–G). Therefore, metformin may reduce the risk of excessive DSC senescence-associated pregnancy loss.

## Discussion

Cellular senescence is an irreversible growth arrest that occurs in response to cellular damage and stress, eventually leading to disrupted tissue homeostasis (Zhang et al, 2022). Cellular oxidative stress, metabolic processes, autophagy, and responses to hypoxia occur during decidualization (Shi et al, 2022; Yu et al, 2021). Here, we found that decidualized ESCs in in vitro models and primary DSCs displayed high levels of cell senescence, which is consistent with recent reports (Brighton et al, 2017; Deryabin et al, 2020; Lucas et al, 2020; Shi et al, 2022). Abnormal senescence of stromal cells reportedly impairs endometrial decidualization and disrupts dialogue with trophoblasts (Deryabin and Borodkina, 2022; Rawlings et al, 2021). This indicates that the presence of a subpopulation of senescent DSCs is an important for the ESC decidualization process. Notably, we found that excessive senescence of DSC was enriched in the decidua of patients with RSA, and mice with D-galactose-induced aging were more prone to pregnancy loss, suggesting that abnormal accumulation of senescent DSCs increases the risk of adverse pregnancy outcomes, including spontaneous abortion. During normal pregnancy, decidualization is accompanied by the appearance of a subpopulation of senescent DSCs; however, during spontaneous abortion, senescent DSCs accumulate excessively in the decidual tissue. This means that homeostasis of senescent DSCs at the maternal-fetal interface is crucial for successful pregnancy. However, the regulatory mechanisms underlying DSC senescence homeostasis during normal pregnancy and excessive senescence in patients with SA have been largely unknown (Wu et al, 2023).

BCAA (including leucine, valine, and isoleucine) play crucial physiological roles in regulating metabolism, protein synthesis, and cellular senescence. Long-term exposure to high-BCAA diets results in obesity, hyperphagia, and reduced lifespan (Kitada et al, 2019; Solon-Biet et al, 2019). The transporters SLC7A5/SLC3A2 facilitate the intake and tissue abundance of BCAAs. Interestingly, we observed that these transporters are upregulated during decidualization, particularly SLC3A2. Furthermore, massive SLC3A2 accumulation and BCAA uptake have been observed in the decidua of patients with RSA. Notably, BCAA deprivation suppressed DSC senescence in vitro, especially leucine deprivation. However, exposure to high-leucine diets leads to increased pregnancy loss in mice and uterine senescence. BCAA play important roles in regulating the mechanistic target of rapamycin (mTOR), insulin, and insulin-like growth factor (IGF)-1 and further link nutrition and metabolism with health and aging (Kitada et al, 2019; Solon-Biet et al, 2019; Soultoukis and Partridge, 2016). Our study, and those of others, have shown that IGF-1 signaling and mTOR-associated autophagy are required for decidualization and normal pregnancy (Lu et al, 2021; Shi et al, 2022; Yang et al, 2018). Therefore, these pathways may be involved in spontaneous abortion induced by the excessive accumulation of leucine. Here, we found that decidualization was also accompanied by activation of the p38 MAPK pathway. Further analysis showed that leucine-induced DSC senescence was dependent on the p38 MAPK signaling pathway. Pathways such as insulin/IGF signaling, mTOR, and NAD+ reportedly affect aging in many organisms (Smith et al, 2020). Therefore, the possible roles of the insulin/IGF and mTOR signaling pathways in leucine-associated DSC senescence and pregnancy loss cannot be ignored and require further research.

At the maternal-fetal interface, dNK cells are the most abundant immune cells during early pregnancy and play essential roles throughout pregnancy. Phenotypically, most human uterine NK cells are characterized as NCAM1bright FCGR3A−ITGA1+ cells, and these cells exhibit lower cytotoxic activity (Kopcow et al, 2005; Male and Moffett, 2023). Generally, senescent cells are cleared by immune cells, including NK cells, and granule exocytosis

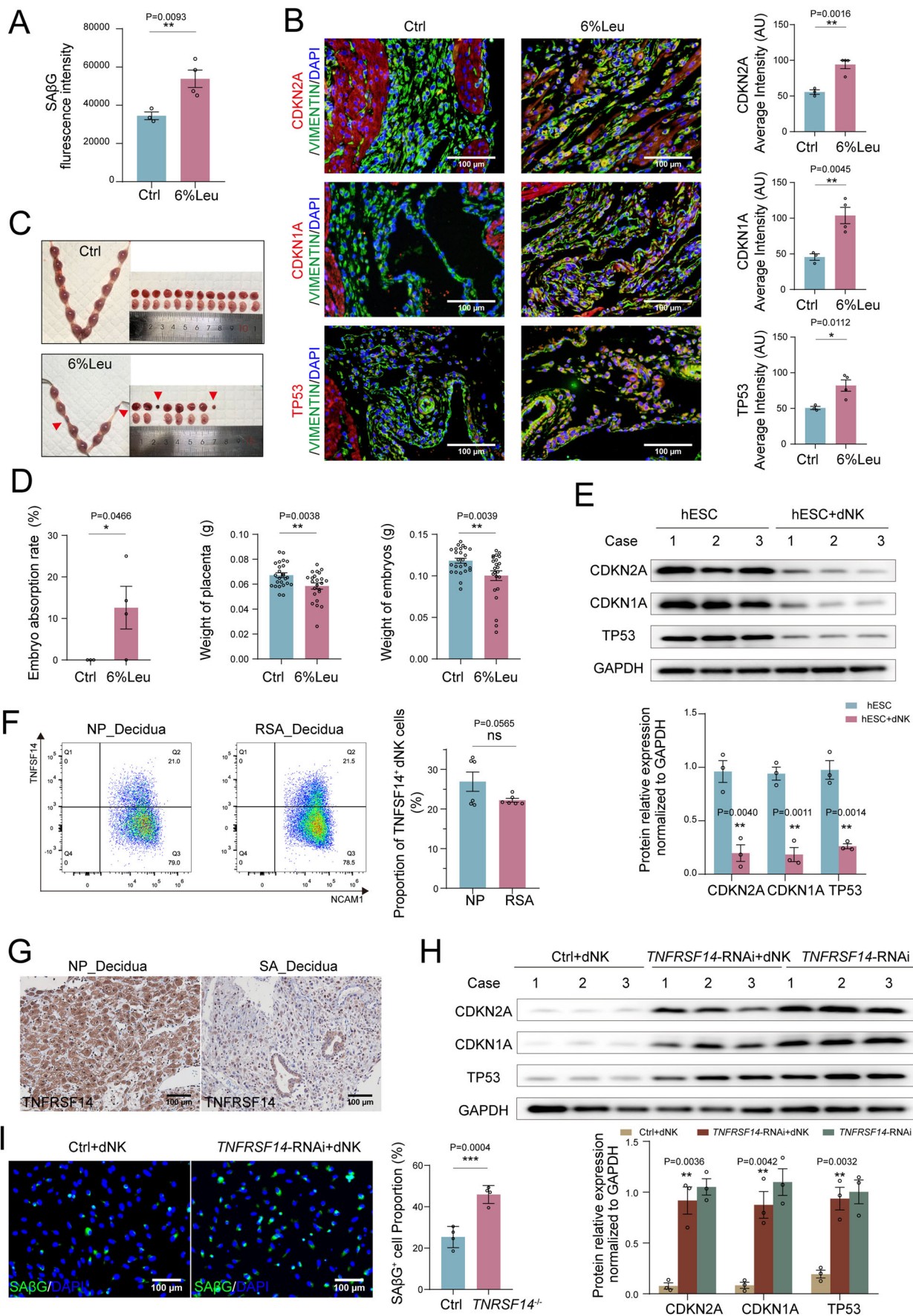

**Figure 4.  dNK cell inhibits DSC senescence in a TNFSF14/TNFRSF14-dependent manner.**

(A–C) C57BL/6 pregnant mice were fed with control (1.2% leucine) ($n = 3$ biological replicates) or high leucine (6%) ($n = 4$ biological replicates) fodder, the SAβG activity of DSCs were measured by flow cytometry, and expression of CDKN2A, CDKN1A, and TP53 of DSCs were detected by immunofluorescence. (D) The embryo resorption rate, the weight of placenta and embryo of pregnant mice were quantified at the gestation of day 13.5 (control fodder: $n = 3$ biological replicates, high leucine fodder: $n = 4$ biological replicates). (E) Decidualized hESCs were co-cultured with dNK cells for another 48 h, and CDKN2A, CDKN1A, and TP53 expression were detected by western blotting ($n = 3$ biological replicates per group), relative expression levels of proteins were standardized using internal reference GAPDH. (F) Compared the expression of TNFSF14 on dNK cells of NP ($n = 6$ biological replicates) and SA ($n = 6$ biological replicates) by flow cytometry. (G) Compared the expression of TNFRSF14 on DSCs of NP ($n = 6$ biological replicates) and SA ($n = 6$ biological replicates) by Immunohistochemistry. Scale bar, 100 μm. (H, I) Decidualized si*TNFRSF14*-hESCs were co-cultured with dNK cells for another 48 h, western blotting indicated CDKN2A, CDKN1A, and TP53 expression ($n = 3$ biological replicates per group), relative expression levels of proteins were standardized using internal reference GAPDH, and immunofluorescence showed SAβG$^+$ cells ($n = 4$ biological replicates per group). NP: normal pregnancy; RSA: recurrent spontaneous abortion. Data were presented as mean ± SEM. *$P < 0.05$, **$P < 0.01$, ***$P < 0.001$, ns, no significance, using a one-tailed (A, B, D), two-tailed, unpaired Student's t test (E, F, H, I). Source data are available online for this figure.

contributes to this process (Jin et al, 2021; Sagiv et al, 2013). A previous study also suggested that IL15-activated NK cells could clear senescent decidual cells (Brighton et al, 2017). However, the mechanisms by which NK cells inhibit stromal cell senescence and maintain senescence homeostasis at the maternal-fetal interface remain unclear. Here, we observed that dNK cells suppress DSC senescence, whereas the NK cell-depleted mouse model displays uterine senescence and a high risk of adverse pregnancy outcomes, including fetal loss and FGR. Further analysis indicated that TNFSF14$^+$ dNK cells participate in the clearance of DSC senescence by binding to their receptor TNFRSF14. More interestingly, the TNFSF14 protein downregulated the expression of SLC3A2 by HSF1 and further restricted BCAA intake in DSCs. However, the exact mechanism of action of TNFSF14 and TNFRSF14 in HSF1 requires further investigation.

In accordance with D-galactose-induced aging in mice, excessive uterine senescence and a high risk of pregnancy loss were also observed in NK cell-depleted and *Tnfrsf14*$^{-/-}$ mice. We also detected abundant accumulation of senescent DSCs in patients with RSA. Therefore, excessive senescence can be considered a risk factor for spontaneous abortion. Remarkably, we found that the TNFSF14 protein alleviated uterine senescence and reduced the risk of pregnancy loss in NK cell-depleted mice. TNFRSF14$^+$ DSC, but not TNFSF14$^+$ NK cells, were aberrantly decreased in the decidua of patients with RSA. Therefore, exploring other strategies for treating senescence-associated spontaneous abortion in DSCs is particularly important.

Metformin is the most widely used oral hypoglycemic drug for type 2 diabetes and has been proven to play a role in combating age-related disorders and improving the health span (Hou et al, 2023; Zeng et al, 2023). In the reproductive system, metformin can reduce the risks of preeclampsia in pregnant females with severe obesity, miscarriages, and preterm delivery in females with polycystic ovary syndrome (Tosti et al, 2023). More importantly, metformin during pregnancy has been proven safe in an increasing number of randomized clinical trials (Brand et al, 2022). In this study, the administration of metformin was shown to exert protective effects against spontaneous abortion in D-galactose-induced aging mice. More importantly, this is the first report to demonstrate the value of metformin in the treatment of DSC senescence-associated spontaneous abortion. A recent study in mice demonstrated that metformin decreases glucose levels by allosteric inhibition of fructose-1-6-bisphosphatase (FBP1) (Hunter et al, 2018). Fructose-1,6-bisphosphate (FBP) is an endogenous intermediate in the glycolytic pathway that is removed by FBP1

activity and the enrichment of FBP is required for normal pregnancy, as it triggers maternal-fetal immune tolerance, decidualization, and trophoblast invasion (Zhou et al, 2022). In addition, a previous study suggested that metformin could attenuate preterm birth in mice by alleviating the premature aging of the decidua (Sun et al, 2018). Furthermore, many studies have suggested that the mechanisms of metformin attenuate cell senescence including reducing ROS production in mitochondrial complex I, regulating insulin and IGF-1 signaling and mTOR signaling, improving DNA repair, and inhibition of NF-κB signal transduction induced by proinflammatory cytokines (Chen et al, 2022; Sunjaya and Sunjaya 2021). DNA repair disorders are closely associated with reproductive senescence and decreased fertility (Sonowal et al, 2023). Therefore, there are multiple pathways by which metformin prevents spontaneous abortion, including decreasing cell senescence and increasing FBP accumulation in the decidua, improving DNA repair ability, and regulating the mTOR signaling process, which all require further in-depth research.

As depicted in Fig. 8H, stromal cell senescence was enriched during decidualization, along with active oxidative stress and BCAA intake. Mechanistically, SLC3A2-mediated BCAA uptake induces DSC senescence by activating the p38 MAPK signaling pathway, whereas TNFSF14$^+$ dNK cells control DSC senescence via the TNFRSF14-SLC3A2/leucine regulatory axis. The overall goal is to maintain homeostasis of DSC senescence, helping establish and maintain normal pregnancy. However, excessive DSC senescence induced by an imbalance of TNFSF14 and the TNFRSF14-SLC3A2/leucine regulatory axis increases the risk of adverse pregnancy outcomes, including SA and FGR.

Overall, our findings suggest that spontaneous abortion can be described as a senescence-related disease. More excitingly, our study implicates metformin as a potential drug to treat DSC senescence and prevent pregnancy loss.

# Methods

## Patients and sample collection

This study was approved by the Human Research Ethics Committee of the Obstetrics and Gynecology Hospital of Fudan University (No. 2018-25-C2), and all the participants provided written informed consent. Decidua ($n = 30$) were obtained from healthy pregnant females who had undergone selective termination in the first trimester of pregnancy for selective termination

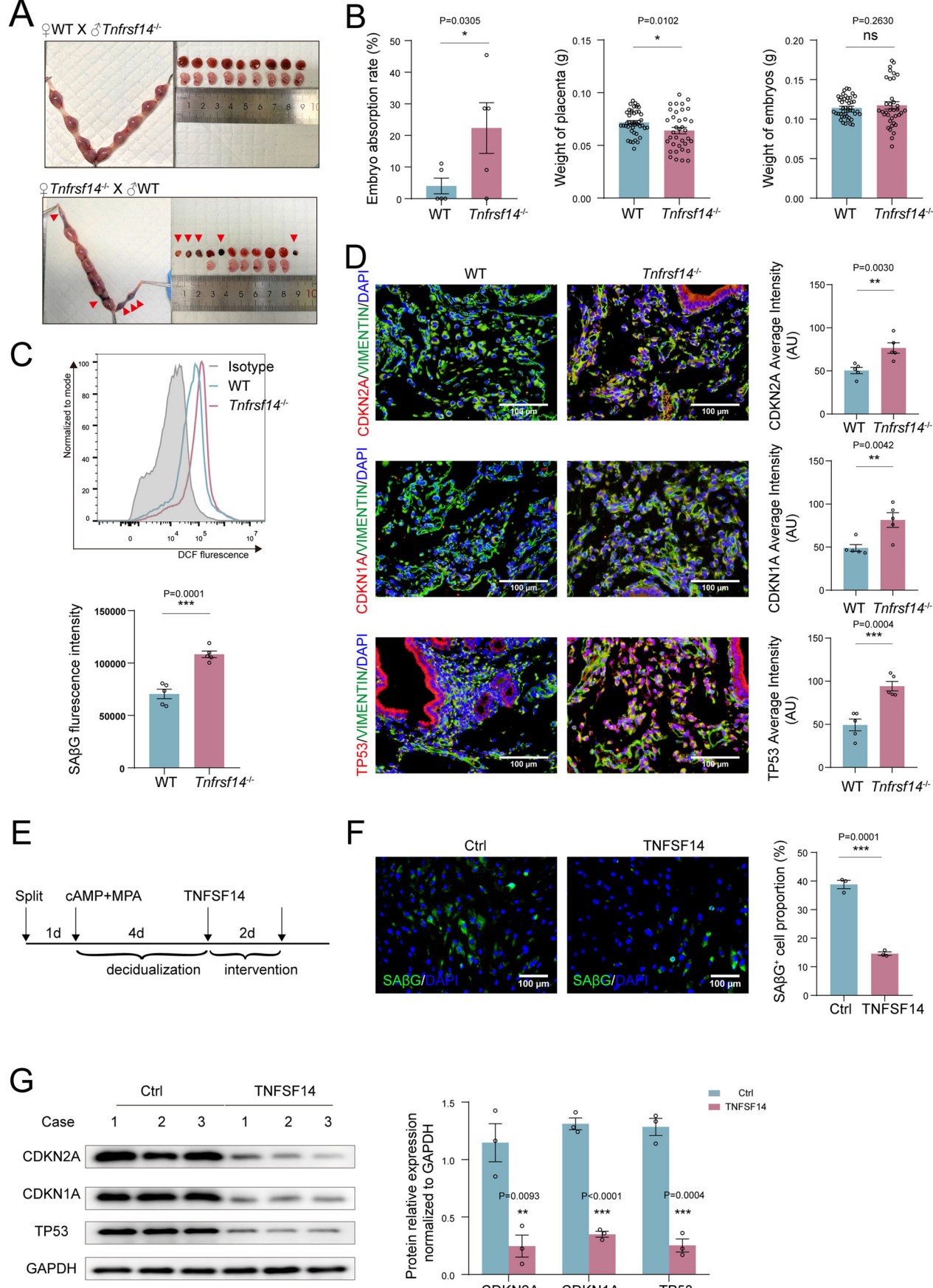

**Figure 5. TNFSF14/TNFRSF14 signal maintains DSC senescence homeostasis and normal pregnancy.**

(A, B) At the gestation of day 13.5, embryo resorption, weight of placenta and embryo in *Tnfrsf14* knockout pregnant mice (*Tnfrsf14*$^{-/-}$♀×WT♂, $n = 5$ biological replicates) and WT pregnant mice (WT♀×*Tnfrsf14*$^{-/-}$♂, $n = 5$ biological replicates) were recorded. (C, D) The SAβG activity of DSCs were measured by flow cytometry, and expression of CDKN2A, CDKN1A and TP53 of DSCs were detected by immunofluorescence ($n = 5$ biological replicates per group). (E) Schematic diagram of decidualized hESCs treated with recombinant TNFSF14 protein. (F, G) After decidualized hESCs were treated with TNFSF14 (250 ng/mL, 48 h), SAβG$^+$ cells were detected by immunofluorescence, and CDKN2A, CDKN1A, and TP53 expression were measured by western blotting ($n = 3$ biological replicates per group), relative expression levels of proteins were standardized using internal reference GAPDH. Scale bar, 100 μm. Data were presented as mean ± SEM. *$P < 0.05$, **$P < 0.01$, ***$P < 0.001$, ns, no significance, using a one-tailed (B–D), two-tailed, unpaired Student's t test (F, G). Source data are available online for this figure.

(gestational age, 7–9 weeks; maternal age, 25–40 years) or from patients ($n = 18$) with unexplained RSA (experiencing two or more consecutive spontaneous miscarriages before the 24th week of pregnancy) caused by non-endocrine or non-genetic factors (gestational age, 7–9 weeks; maternal age, 25–35 years). All cases were histologically verified based on established criteria. Pregnancies were confirmed using ultrasonography and blood tests. Females who had experienced spontaneous miscarriages attributed to endocrine, anatomical, or genetic issues, or due to any infection, were excluded from the study.

The endometria ($n = 18$) were collected from females of reproductive age with a normal pregnancy history (25–40 years old) who underwent diagnostic curettage or hysterectomy for benign reasons unrelated to endometrial dysfunction. None of the patients had received hormone medication within 6 months prior to surgery. All tissues were histologically confirmed according to established standards and were obtained in the secretory phase. Samples were collected under sterile conditions and transported to the laboratory within 30 min of surgery in DMEM/F12 (SH30023.01B; HyClone, Logan, UT, USA) containing 10% fetal bovine serum (FBS; SH30068.03; Hyclone, US origin, heat-inactivated) for further isolation of ESCs, DSCs, and dNK cells.

## Mice

All mice were housed in SPF grade animal rooms. This study was approved by the Experimental Animal Management and Use Committee of Shanghai Model Organisms Co., Ltd (SRCMO-IACUC.NO.2021-0027). C57BL/6 mice (4–8 weeks old) were purchased from Shanghai SLAC Experimental Animal Co., Ltd. (Shanghai, China). After 1 week of adaptive feeding, we performed further experiments. The aging model was constructed by subcutaneous injection of D-galactose (HY-N0210, MedChem Express, Monmouth Junction, NJ, USA) at a dose of 120 mg/kg/d for 8 weeks in accordance with a method used in a previous study (Liao et al, 2016; Liu et al, 2021). The control group was subcutaneously injected with the same dose of stroke-physiological saline solution (SPSS). To rule out the effect of D-galactose on the ovaries, we performed embryo transfer. At 8 weeks of age, female mice were mated with ligated C57BL/6 male mice (8 weeks old). The date of vaginal hydrant detection was recorded as day 0.5 of gestation. Pseudopregnant female mice were used for subsequent uterine transplantation of blastocysts on the day 2.5 of gestation. Meanwhile, we used normal pregnant female mice to provide fertilized eggs. Four-week-old C57BL/6 female mice were intraperitoneally injected with 5 IU of pregnant mare serum gonadotropin, followed by 5 IU of human chorionic gonadotropin 48 h later. At night, the mice were mated with 8-week-old males. On the day of vaginal hydrant detection, mice were euthanized, the ovaries

were separated, and fertilized eggs were isolated. The cleaned fertilized eggs were placed in M2 cell culture medium and transferred to a cell culture incubator for culturing for 4 d, during which the fertilized eggs developed into blastocysts. The blastocysts were then transferred into the uterus of pseudopregnant mice, and seven blastocysts were transferred into each side of the uterus. On the 9th day after blastocyst transplantation (day 13.5 of fertilized egg development), the mice were euthanized to collect the uteri to record pregnancy outcomes. Part of the uterus was fixed for subsequent experiments, and the remaining part was cut and digested into single cells for cytometry. In addition, some D-galactose-induced female mice were treated with metformin (200 mg/kg, every other day; 1115-70-4; Sigma-Aldrich, USA) starting from mating and vaginal hydrant detection. The other embryo transfer methods were consistent with those described above.

C57BL/6 female mice were mated with C57BL/6 male mice (2:1), and the pregnant mice were then fed control (containing 1.2% leucine) or high leucine fodder (containing 6% leucine). On day 13.5 of gestation, the uteri were excised for further analysis. The fodder was purchased from Wuxi Fanbo Biotechnology Co., Ltd. (Wuxi, China).

To remove NK cells from the bodies of pregnant mice, they were intraperitoneally injected with an anti-NK1.1 neutralizing antibody (15 mg/kg, every other day; BioLegend, 108760). Control mice were injected with the same amount of isotype control antibody. In addition, to investigate the role of TNFSF14, we treated NK cell-depleted pregnant mice with intraperitoneal injection of TNFSF14 recombinant protein (10 μg, every 2 d; HY-P73837; MedChem Express). This indicates that the NK1.1 neutralizing antibody was injected on the day 0 of pregnancy and then injected every 2 d. The injections were administered on days 0, 2, 4, 6, 8, 10, and 12 of pregnancy. We observed the pregnancy outcomes on day 13.5 of gestation.

*Tnfrsf14* knockout mice (cat. NO. NM-KO-190170) were constructed by Shanghai Model Organisms Center, Inc. (Shanghai, China). The *Tnfrsf14* knockout mice were generated by knocking out the exon 2 region of the *Tnfrsf14* gene. To rule out the influence of the embryonic genotype, female *Tnfrsf14* knockout (*Tnfrsf14*$^{-/-}$) mice were mated with male wild-type mice as the experimental group, and female wild-type mice were mated with male *Tnfrsf14*$^{-/-}$ mice as the control group. From day 0 of pregnancy, *Tnfrsf14*$^{-/-}$ pregnant mice were administered metformin (200 mg/kg every other day) by gavage.

## Single-cell sequencing

Raw single-cell sequencing data were obtained from the NCBI Gene Expression Omnibus (GEO) database. The datasets were GSE183837 (endometrium) and GSE194219 (decidua). scRNA-seq data analysis

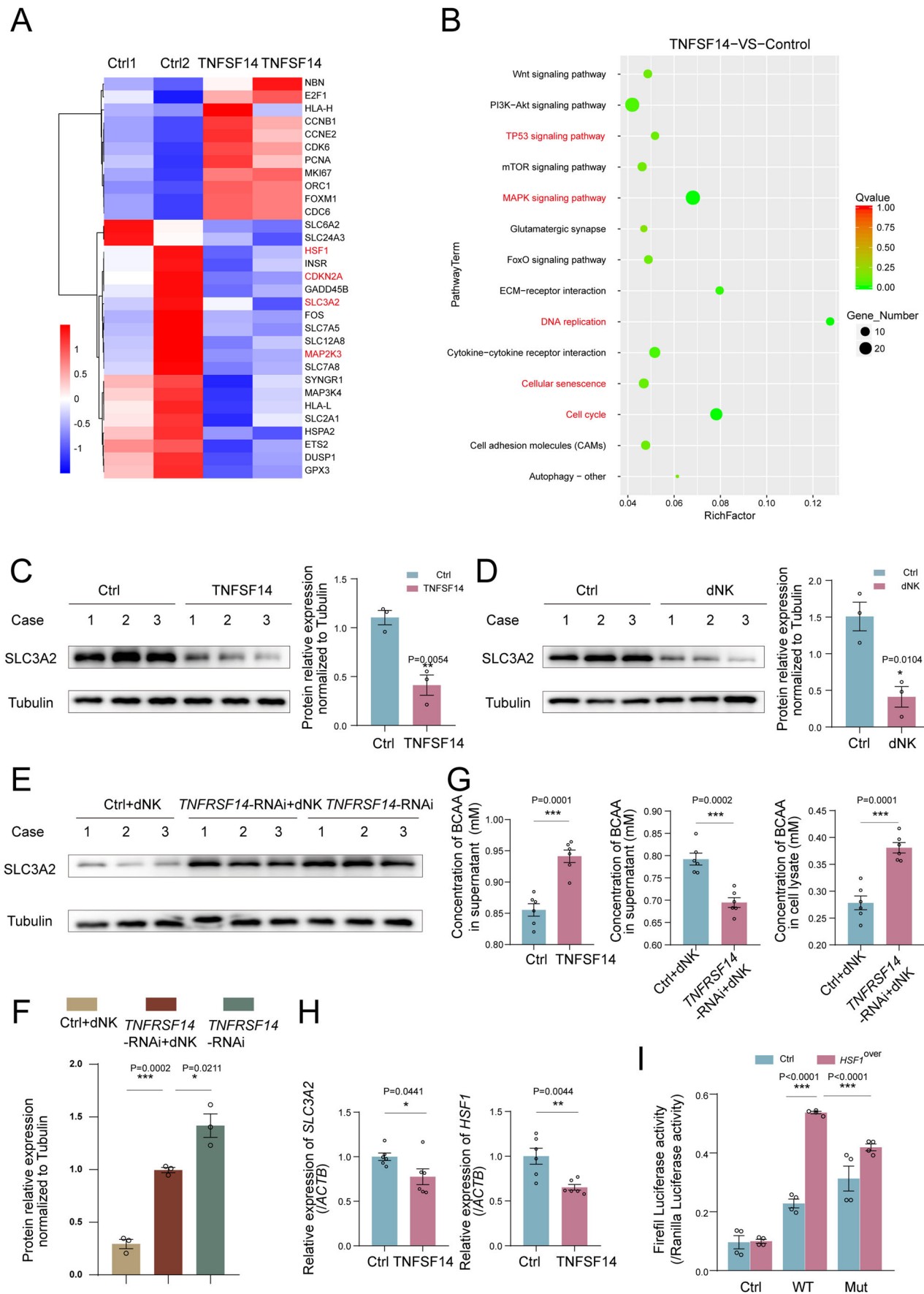

**Figure 6. TNFSF14⁺ dNK cell control DSC senescence by the SLC3A2/BCAA signaling pathway.**

(A) Bubble diagram showed the enrichment of pathway for DSCs treated with TNFSF14. (B) Heat map showed the relative expression of differential gene for DSCs treated with TNFSF14. (C, D) Decidualized hESCs were treated with TNFSF14 (250 ng/mL), or co-cultured with dNK cells (5 × 10⁵) for 48 h, SLC3A2 expression levels were detected with western blotting (n = 3 biological replicates per group), relative expression levels of proteins were standardized using internal reference Tubulin. (E, F) Decidualized siTNFRSF14 hESCs were co-cultured with dNK cells for 48 h, SLC3A2 expression were detected by western blotting (n = 3 biological replicates per group), relative expression levels of proteins were standardized using internal reference Tubulin. (G) Decidualized hESCs were treated with TNFSF14 (250 ng/mL), or decidualized siTNFRSF14 hESCs were co-cultured with dNK cells for 48 h, and then the concentration of BCAA was measured (n = 6 biological replicates per group). (H) After decidualized hESCs were treated with TNFSF14 (250 ng/mL, 48 h), the transcriptional levels of SLC3A2 and HSF1 were detected by RT-PCR (n = 6 biological replicates per group). (I) Dual luciferase reporter assays were conducted in hESCs to verify the combination of HSF1 and WT or mutated SLC3A2 promoter region (n = 4 biological replicates per group). Data were presented as mean ± SEM. *P < 0.05, **P < 0.01, ***P < 0.001, using a two-tailed, unpaired Student's t test. Source data are available online for this figure.

was conducted by NovelBio Bio-Pharm Technology Co., Ltd. Single-cell dissociation and sequencing were performed as described in our previous study (Shi et al, 2022). QuSAGE analysis was used to describe the relative pathway activation of specific genes in stromal cells of the endometrium and decidua. These results suggest that the senescence pathway was enriched. Cell-related genes (cell cycle-related genes: CDKN2A, CDKN1A, TP53, CDK6, and RB1; SASP-related genes: IL1B, IL15, TIMP1, EGFR, etc.) were identified among ESCs and DSCs using FindMarkers with the Wilcoxon rank-sum test algorithm. To explore the specific mechanism of stromal cell senescence, functional enrichment analysis was performed using Gene Ontology (http://www.geneontology.org) and each enriched ontology hierarchy (false discovery rate [FDR] < 0.05) was reported with two terms in the hierarchy. Differential gene expression analysis (including cell senescence-related genes; BCAA transport: SLC3A2, SLC7A5, and SLC1A5; oxidative stress genes: GPX1, GPX4, and P4HB) was performed using FindMarkers with the Wilcoxon rank-sum test algorithm.

## Cell isolation

Primary ESCs were obtained from the secretory phase of the normal endometrium, DSCs and dNK cells were isolated from the decidua as performed previously (Shi et al, 2022). The tissues were washed with PBS (SH30256.01; HyClone) to remove any remaining blood. The endometria were cut and digested with 20% collagenase type IV (9003–98-9, Sigma-Aldrich) for 20 min at 37 °C. The supernatant was sieved through a 70 μm Falcon cell strainer (Corning, Corning, NY, USA), and centrifuged at 1500 rpm for 8 min. The supernatant was discarded and 15 mL of red blood cell lysis buffer (R1010, Solarbio, China) was added for 15 min on ice to remove erythrocytes. After washing with PBS, the cell pellets were resuspended in DMEM/F12 containing 10% charcoal/dextran-treated FBS (SH30068.03; Hyclone, US origin, heat-inactivated) and 1% penicillin/streptomycin/neomycin solution (Sangon Biotech, Shanghai, China) and incubated in culture flasks to allow the ESCs to adhere overnight. The culture medium was replaced every 2 d.

The decidua was sectioned and digested with 20% collagenase type IV for 30 min at 37 °C. The samples were sieved through sterile gauze pads (pore diameter sizes: 100, 200, and 400 mesh), and the suspension was centrifuged at 1500 rpm for 8 min. The cells were then suspended in DMEM/F12, slowly added to a discontinuous gradient of 20%, 40%, and 60% Percoll bulk standard, and centrifuged at 2500 rpm for 30 min. The DSCs were at the 20%/40% interface and the DICs were at the 40%/60% interface. DSCs

and DICs were cultured overnight in a culture flask to remove the other cells. The DSC culture medium was replaced every 2 d. DICs were collected to isolate dNK cells.

Decidual NK cells were isolated using MACS (Human NK Cell Isolation kit, 130–092-601, Miltenyi Biotec, Bergisch Gladbach, Germany). The number of obtained DICs were counted. Next, 10 μL biotin-antibody cocktail and 40 μL MACS buffer were added to every 10⁷ DICs. This was mixed well and left to incubate at 4 °C in the dark for 7 min. Then, 20 μL of MicroBear cocktail and 30 μL MACS buffer were added to each 10⁷ DIC sample, which was mixed well and left to incubate at 4 °C in the dark for 12 min. The cell suspension was slowly added to the MS column and the leaked cell suspension was collected. The obtained cell suspension was centrifuged at 1500 rpm for 5 min, and the supernatant was discarded to obtain the NK cells. Flow cytometry was used to verify the purity of NK cells.

The purity of the DSCs and ESCs was greater than 95%, and that of the NK cells was greater than 90%, as determined by flow cytometry.

## Cell treatment

hESCs were obtained from the American Type Culture Collection (CRL-4003; ATCC, Manassas, VA, USA), and the passage number for all hESC lines was limited to no more than six generations. hESCs and primary ESCs were treated with cAMP (0.5 mM; HY-12306; MedChem Express) plus MPA (1 μM; HY-B0648, Med-Chem Express) for different time gradients. In addition, during decidualization, hESCs were cultured in a control medium, and those lacking all BCAA, leucine, isoleucine, and valine. The culture media was provided by Shanghai HaRO Life Co., Ltd. The hESCs were also treated with PGE₂ (1 μM; HY-101952, MedChem Express) plus MPA (1 μM), as in a previous study (Stadtmauer and Wagner, 2022).

To investigate the effects of LIGHT, decidualized hESCs or DSCs were treated with vehicle or human recombinant TNFSF14 (250 ng/mL; DQU0119011, R&D Systems, Minneapolis, MN, USA) for 48 h. Decidualized hESCs were co-cultured with dNK cells for 48 h to explore the role of NK cells. The DCSs isolated from aborted decidual tissue were treated with metformin (2.5 mM) for 48 h.

## Cell transfection

siRNAs targeting TNFRSF14 (siTNFRSF14) and SLC3A2 (siSLC3A2), as well as the control lentivirus were provided by Shanghai Genechem

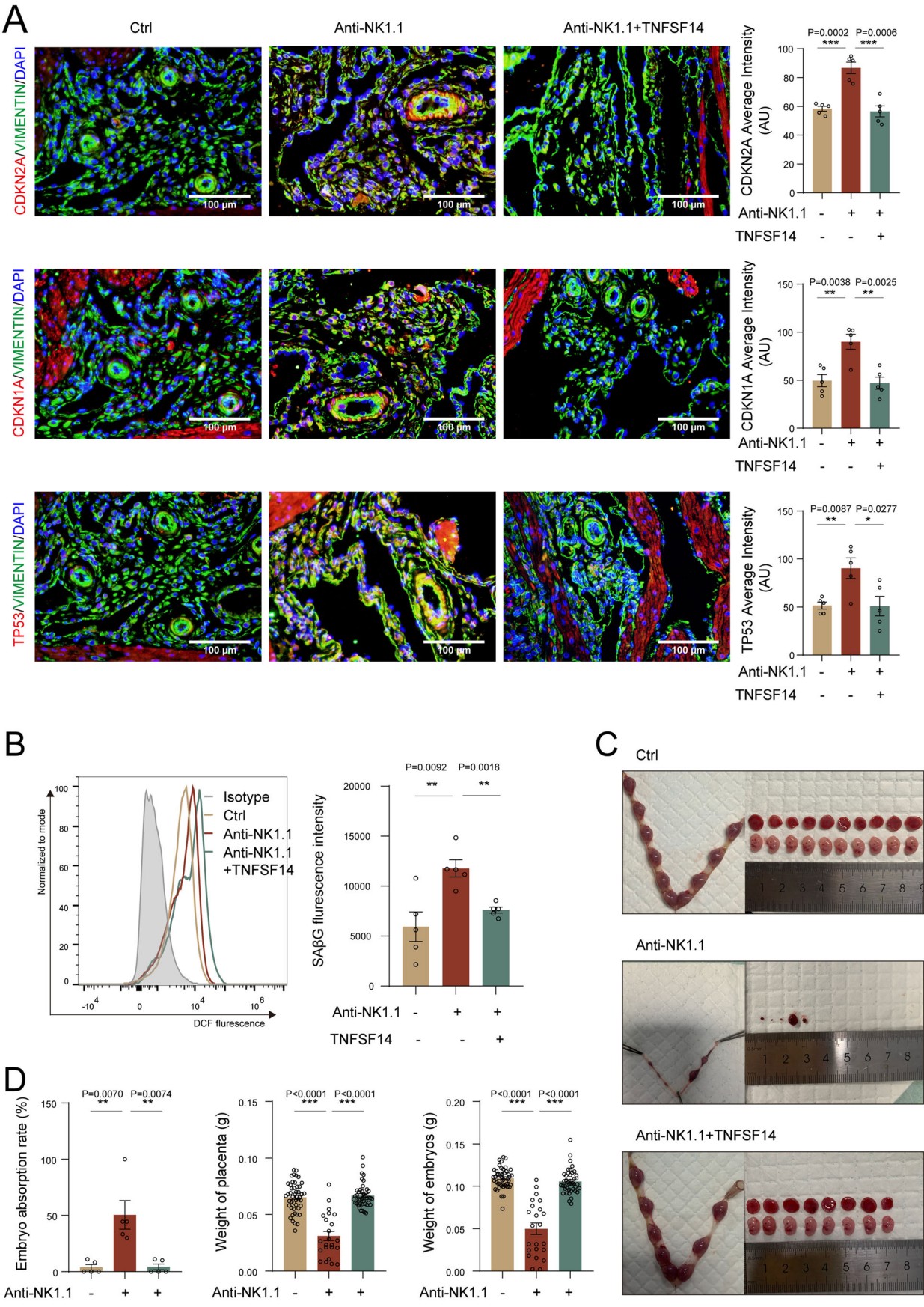

◄ **Figure 7.  TNFSF14 reduces risk of NK cell depletion-associated pregnancy loss with excessive DSC senescence.**

After C57BL/6 pregnant mice were treated with NK1.1 neutralizing antibody ($n = 5$ biological replicates) or isotype control IgG antibody ($n = 5$ biological replicates) by intraperitoneal injection, and at the same time, mice were injected with recombinant TNFSF14 protein intraperitoneally ($n = 5$ biological replicates). (A, B) SAβG activity of DSCs were measured by flow cytometry, and CDKN2A, CDKN1A, and TP53 expression of DSCs were verified by immunofluorescence. (C, D) The pregnancy outcome, embryo resorption, weight of placenta, and embryo were counted at the gestation of day 13.5. Data were presented as mean ± SEM. *$P < 0.05$, **$P < 0.01$, ***$P < 0.01$, using a two-tailed, unpaired Student's t test. Source data are available online for this figure.

Co., Ltd. (Shanghai, China). hESCs were seeded in culture plates and subsequently infected with si*TNFRSF14*, si*SLC3A2* or control lentivirus at 50% confluency. After 12 h of infection, complete culture medium was replaced with fresh medium. Infection efficiency was determined using RT-PCR at 96 h after infection.

The target sequences were as follows: si*TNFRSF14:* GAC CAATTGGCCTAATCAT and si*SLC3A2:* GCTGGGTCCAAT TCACAAGAA.

## Dual luciferase reporter assays

*HSF1* overexpressing mimics, *SLC3A2* and *SLC3A2* mutant luciferase reporter plasmids were generated by Shanghai Genechem Co., Ltd. (Shanghai, China). The plasmids were transfected into decidual cells using Lipofectamine™ 3000 (L3000015; Thermo Fisher Scientific, Waltham, MA, USA). A Dual-Luciferase® 561 Reporter (DLR™) Assay (e1910; Promega, Madison, WI, USA) was used to analyze luciferase activity.

## Real time (RT)-PCR

RNA was extracted using TRIzol reagent (Takara, Kusatsu, Japan). A NanoDrop Technologies spectrophotometer (Thermo Fisher Scientific) was used to quantify the concentration and purity of RNA. RNA was reverse transcribed into cDNA using the Prime-Script RT Reagent Kit (Takara). RT-PCR was conducted using SYBR Green PCR Master Mix (Yeasen Biotechnology Co., Ltd., Shanghai, China) and analyzed using an ABI Prism 7900 Fast Sequence Detection system (Thermo Fisher Scientific). Target mRNA expression was normalized to *ACTB* expression. The results were analyzed using the $2^{-\Delta\Delta Ct}$ method. All primers used for RT-PCR are listed in Table S1.

## RNA-seq

TRIzol reagent was used to extract total RNA. The cDNA library constructed from pooled RNA was sequenced using an Illumina 2000/ 2500 (San Diego, CA, USA) sequencing platform. Each sample mapped read was collected using StringTie. All transcriptomes of the samples were then combined, and Perl scripts were used to reconstruct a comprehensive transcriptome. After generating the final transcriptome, StringTie and Ballgown were used to estimate transcript expression. StringTie is used to perform mRNA expression through calculating fragments per kilobase of transcript per million mapped reads (FPKM). The differentially expressed mRNAs and genes were selected with $\log_2$ (fold change) $>1$ or $\log_2$ (fold change) $<-1$ and with statistical significance ($P < 0.05$) as assessed using R. Differential gene function analysis was performed using DESeq2, while ggplot2 was used for signaling pathway enrichment analysis, followed by graphical display of the enrichment analysis results. According to the pathway

enrichment results, the differential gene expression abundance of the pathway was listed using a heatmap among the control and TNFSF14-treatment groups.

## Immunohistochemistry

Paraffin sections of samples were baked at 60 °C for 2 h, deparaffinized with dimethylbenzene, and then rehydrated with an ethanol series. Tissue sections were boiled in Tris–EDTA buffer (pH 9.0) (C1038; Solarbio, Beijing, China) or Sodium Citrate Antigen Retrieval Solution (C1032; Solarbio) for 15 min to perform antigen retrieval. After cooling to room temperature, the endogenous peroxidase was removed using 3% hydrogen peroxide. The sections were incubated with 5% BSA for 1 h at room temperature. The samples were incubated with rabbit anti-CDKN2A antibodies (1:100; human: ab108349; mouse: ab189034; Abcam), rabbit anti-CDKN1A (1:100; human: ab109520; mouse: ab188224; Abcam), rabbit anti-TP53 (1:100; human: ab32389; mouse: ab131442; Abcam), rabbit anti-TNFRSF14 (1:100; ab185711; Abcam), or rabbit IgG isotypes at 4 °C overnight in humid chamber. After washing three times with PBS, the sections were incubated with a horseradish peroxidase-labeled secondary antibody at room temperature for 30 min. DAB was used to stain the samples and hematoxylin was used to label the nuclei.

## Immunofluorescence

Baking, dewaxing, rehydration, antigen retrieval, and serum blockade were performed as described above. The samples were incubated with primary antibodies overnight at 4 °C, then incubated with secondary antibodies for 2 h at room temperature in the humid chamber. DAPI (C0065; Solarbio) was used to label the nuclei. The dilutions of the antibodies were as follows: CDKN2A (1:100; human: ab108349; mouse: ab189034; Abcam), CDKN1A (1:100; human: ab109520; mouse: ab188224; Abcam), TP53 (1:100; human: ab32389; mouse: ab131442; Abcam), SLC3A2 (1:100; ab244356; Abcam), vimentin (1:100; ab8978; Abcam), and secondary goat anti-mouse (1:1000; ab150113; Abcam) and goat anti-rabbit (1:1000; ab150080; Abcam) antibodies.

## Western blotting

Cells were lysed in RIPA (P0013B, Beyotime) including 1% phenylmethanesulfonylfluoride (PMSF; ST506, Beyotime) proteinase, detached with a cell scraper, and centrifuged for 30 min at $12,000 \times g$. BCA protein assay kit (P0012, Beyotime) used to quantify protein concentrations. Cell lysates were boiled at 99 °C for 10 min for conservation. Protein (20 µg) was electrophoresed in SDS-PAGE gels (PG113, Epizyme Biotech) using a Miniprotein III system (1658033, Bio-Rad), transferred to PVDF membranes

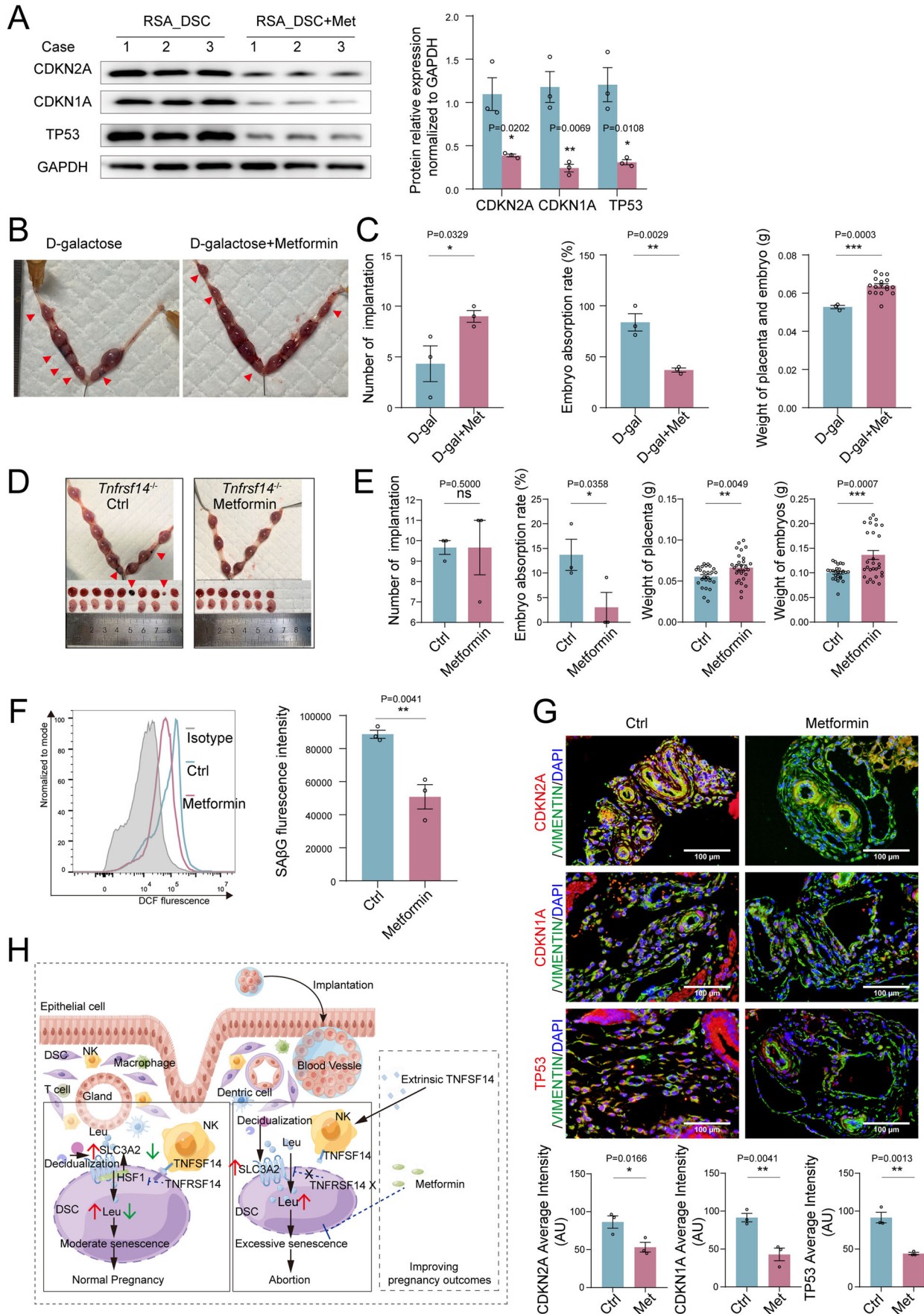

◄ **Figure 8. Metformin reduces the risk of excessive DSC senescence-associated pregnancy loss.**

(A) DSCs of spontaneous abortion were treated with metformin (2.5 mM) for 48 h, CDKN2A, CDKN1A, and TP53 expression were detected by western blotting ($n = 3$ biological replicates per group), relative expression levels of proteins were standardized using internal reference GAPDH. (B, C) After treatment with metformin (200 mg/kg, every other day) in aging pregnant female mice, number of blastocyst implantation, embryo absorption, weight of placenta and embryo were counted at the gestation of day 13.5 (D-galactose group: $n = 3$ biological replicates, metformin group: $n = 3$ biological replicates). (D, E) $Tnfrsf14^{-/-}$ pregnant mice were treated with metformin by gavage ($n = 3$, biological replicates per group, 200 mg/kg, every other day), embryo resorption, weight of placenta and embryo were calculated at the gestation of day 13.5. (F, G) SAβG activity of DSCs were measured by flow cytometry, and CDKN2A, CDKN1A, and TP53 expression of DSCs were verified by immunofluorescence ($n = 3$ biological replicates per group). Scale bar, 100 μm. (H) Schematic roles of decidual NK cells in preventing spontaneous abortion by maintaining homeostasis of DSC senescence during early pregnancy. During decidualization, increased BCAA transport mediated by SLC3A2 induces senescent DSCs, whereas TNFSF14+ dNK cells control DSC senescence by the TNFSF14/TNFRSF14-SLC3A2/leucine regulatory axis. Excessive senescence of DSC induced by the imbalance of TNFSF14/TNFRSF14-SLC3A2/leucine regulatory axis increases the risk of adverse pregnancy outcomes. The image was drew by the Figdraw. Data were presented as mean ± SEM. *$P < 0.05$, **$P < 0.01$, ***$P < 0.001$ using a one-tailed (C, E–G), two-tailed, unpaired Student's t test (A). Source data are available online for this figure.

(ISEQ00010, Millipore) for 70 min, and then incubated in 5% skimmed milk in TBST (P0222, Beyotime) for 1 h at room temperature. PVDF were incubated with primary antibody overnight at 4 °C. Then PVDF were washed with TBST three times, and incubated at room temperature for 1 h in peroxidase-conjugated goat anti-rabbit IgG secondary antibody (1:10,000, BS10003, Bioworld Technology). After washed with TBST, membranes was for chemiluminescence using the Immobilon Western Chemiluminescent HRP Substrate Kit (WBKLS0100, Millipore). The primary antibody are as follows: CDKN2A (1:1000, ab108349, Abcam), CDKN1A (1:1000, ab109520, Abcam), TP53 (1:1000, ab32389, Abcam), IGFBP1 (1:1000, ab181141, Abcam), SLC3A2 (1:1000, ab244356, Abcam), MAP2K3(1:1000, ab195037, Abcam), P38 (1:1000, 9212, Cell Signaling Technology), PP38 (1:1000, ab195049, Abcam), TNFRSF14 (1:500, Abcam, ab185711), Tubulin (1:2000, Bioworld Technology, AP0064), and GAPDH (1:3000, AF0911, Affinity). And Image J was used to calculate the band density. The relative expression levels of proteins were standardized using internal reference GAPDH or Tubulin.

## SA-β-gal staining

SA-β-gal activity was detected in cells using the SPiDER-β-Gal reagent (SG03; DOJINDO LABORATORIES, Kamimashiki, Japan) according to the manufacturer's instructions. Cells were incubated with Bafilomycin A1 for 1 h to inhibit the endogenous β-activity of galactosidase, and then incubated with SPiDER-β-Gal and Hoechst 33342 for 30 min at 37 °C. The cells were observed under a fluorescence microscope after washing with Hanks' balanced salt solution.

## Flow cytometry assays

The flow cytometric antibodies used are listed in Table S2. All flow cytometry assays were performed according to the manufacturer's instructions. On day 13.5th pregnancy, the mice were euthanized and the embryos and placenta were dissected from the uterus. The uterus was then cut into small pieces and 50% type IV collagenase was added for 1 h of digestion. The cell suspension was filtered through a 70 μm filter and centrifuged at 1500 rpm for 8 min. The supernatant was then discarded, 10 mL of red blood cell lysis solution was added and placed on ice for 10 min, before 20 mL of PBS was added to terminate the reaction. This was then centrifuged at 1500 rpm for 8 min to obtain the cell precipitate. Cell membrane surface molecule flow cytometry antibodies were added, and the cells were incubated at room temperature in the dark for 30 min.

The cells were washed with PBS, centrifuged, and the supernatant was discarded. The permeabilization agent was added and left to incubate at 4 °C in the dark for 40 min. The cells were washed with PBS, centrifuged, and the supernatant was discarded. Intracellular antibodies were added, and the cells were incubated at room temperature in the dark for 30 min. These were washed with PBS and the supernatant was discarded, before 200 μL PBS was added to resuspend the cells. Samples were analyzed using a CytoFLEX flow cytometer (Beckman Coulter, Inc., Brea, CA, USA) and FlowJo software was used to analyze the data.

## BCAA measurement

The measurement of BCAA in cells, supernatant and tissue used the Branched Chain Amino Acid Assay Kit (abcam, ab83374). Tissue (40 mg) or cells ($4 \times 10^6$) were homogenized with 200 μL assay buffer. And then centrifuged at $15,000 \times g$ for 10 min to remove cell debris and other insoluble materials, and 20 μL of supernatant used to measure BCAA levels. The concentrations of BCAA were measured according to the manufacture's protocol.

## Reactive oxygen species detection

The ROS were detected by a 2,7-dichlorofluorescin diacetate (DCFH-DA) assay kit (S0033S, Beyotime). Each sample was incubated with DCFH-DA (10 μmol/L) for 20 min at 37 °C, and rinsed with PBS. Flow cytometry measured the ROS levels in cells.

## Statistics analyses

All the data were analyzed by STATA (version 15, USA), and presented as mean ± SEM. Student's t test was used to analyze between two groups, and one-way ANOVA test was conducted among multiple groups. The differences were considered as statistically significant at $P < 0.05$.

# Data availability

Single-cell sequencing: GSE183837 (endometrium, https://www.ncbi.nlm.nih.gov/geo/query/acc.cgi?acc=GSE183837) and GSE194219 (decidua, https://www.ncbi.nlm.nih.gov/geo/query/acc.cgi?acc=GSE194219). RNA sequencing: PRJNA1061093 (Sequence Read Archive, https://dataview.ncbi.nlm.nih.gov/object/PRJNA1061093?reviewer=bc9lc66o0o117ft6d7ej8te74j).

The source data of this paper are collected in the following database record: biostudies:S-SCDT-10_1038-S44318-024-00220-3.

## Peer review information

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

## Acknowledgements

Present research was supported by the National Key Research and Development Program of China (2023YFC2705403), the Major Research Program of National Natural Science Foundation of China (NSFC 32370914, 92357306, 81901563, 82301895), the Shanghai Oriental Talent Plan, the Shanghai Natural Science Foundation (23ZR1408200), Innovative research team of high-level local universities in Shanghai (SHSMU-ZDCX20211100), and the Shanghai Sailing Program (22YF1404000, 19YF1438500).

## Author contributions

**Jia-Wei Shi**: Conceptualization; Resources; Data curation; Software; Formal analysis; Validation; Investigation; Visualization; Writing—original draft. **Zhen-Zhen Lai**: Resources; Data curation; Validation; Investigation; Methodology; Writing—review and editing. **Wen-Jie Zhou**: Resources; Funding acquisition; Methodology; Writing—review and editing. **Hui-Li Yang**: Resources; Software; Formal analysis; Funding acquisition; Methodology; Writing—review and editing. **Tao Zhang**: Resources; Software; Methodology; Writing—review and editing. **Jian-Song Sun**: Resources; Software; Methodology; Writing—review and editing. **Jian-Yuan Zhao**: Conceptualization; Resources; Software; Supervision; Methodology; Project administration; Writing—review and editing. **Ming-Qing Li**: Conceptualization; Resources; Software; Supervision; Funding acquisition; Methodology; Project administration; Writing—review and editing.

Source data underlying figure panels in this paper may have individual authorship assigned. Where available, figure panel/source data authorship is listed in the following database record: biostudies:S-SCDT-10_1038-S44318-024-00220-3.

## Disclosure and competing interests statement

The authors declare no competing interests.

