## [Peer Review File · The EMBO Journal]

TNFSF14+ natural killer cells prevent spontaneous abortion by restricting leucine-mediated decidual stromal cell senescence

Jia-Wei Shi, Zhen-Zhen Lai, Wen-Jie Zhou, Hui-Li Yang, Tao Zhang, Jian-Song Sun, Jian-Yuan Zhao, and Ming-Qing Li

Corresponding authors: Ming-Qing Li (mqli@sjtu.edu.cn) , Jian-Yuan Zhao (zhaojianyuan1508@xinhumed.com.cn)

Review Timeline:

Submission Date:	4th Jan 24
Editorial Decision:	15th Feb 24
Revision Received:	13th Jun 24
Editorial Decision:	24th Jul 24
Revision Received:	31st Jul 24
Accepted:	4th Aug 24

Editor: Kelly Anderson

Transaction Report:

Dear Prof. Li,

Thank you for submitting your manuscript for consideration by the EMBO Journal. It has now been seen by three referees whose comments are shown below.

Given the referees' recommendations, I would like to invite you to submit a revised version of the manuscript, addressing the comments of all three reviewers. In particular in this case the English language writing quality will also need improvement. If you need recommendations for professional services, please let me know. I should add that it is EMBO Journal policy to allow only a single round of revision, and acceptance of your manuscript will therefore depend on the completeness of your responses in this revised version. It would be good to discuss your plan to address the referee concerns, and I am available to do so by email or zoom in the coming weeks.

Thank you for the opportunity to consider your work for publication. I look forward to your revision.

Yours sincerely,

Kelly M Anderson, PhD
Editor, The EMBO Journal
k.anderson@embojournal.org

Please remember: Digital image enhancement is acceptable practice, as long as it accurately represents the original data and conforms to community standards. If a figure has been subjected to significant electronic manipulation, this must be noted in the figure legend or in the 'Materials and Methods' section. The editors reserve the right to request original versions of figures and

the original images that were used to assemble the figure.

We realize that it is difficult to revise to a specific deadline. In the interest of protecting the conceptual advance provided by the work, we recommend a revision within 3 months (15th May 2024). Please discuss the revision progress ahead of this time with the editor if you require more time to complete the revisions.

Referee #1:

The authors have used secondary and primary cells, murine and human ex vivo models to study cellular senescence in the endometrium. They have identified one mechanisms of cellular senescence by studying the BCAA transporter SLC3A2, and recognized that uNK cells regulate senescence through the LIGHT-HVEM interactions. A lot of work characterizing senescence was performed comparing control patients to those with a history of spontaneous abortions, implicating these mechanistic pathways in reproductive outcomes.

Whilst the work is performed to a high standard and is topical in the field of endometrial senescence, I feel the authors need to address the major points raised below in order to substantiate their conclusions.

Major concerns:

1. The laboratory work investigating the cellular and ex vivo studies is of high quality, and the murine model adds to the study. However, a large portion of the work relies on comparing human control endometrial biopsies with those from patients with unexplained spontaneous abortions (SA). The methods and source of endometrial biopsies is unclear.

More clarity is needed in the patient and sampling collection of the methods.

- How has the SA group been diagnosed? Do they suffer from recurrent SA? If so, how many? Do you have information on how long ago the spontaneous abortions occurred?
- How were the biopsies for cultures/uNKs timed to the secretory phase? Urinary LH? Patient-reported cycling? Senescence and uNK phenotype changes dramatically over the secretory phase so authors need to ensure fair comparisons. How can the authors be sure that changes were not temporal?

2. The authors are using pregnant endometrium to compare control and SA, but at no point in the manuscript is this acknowledged. Instead, conclusions are extrapolated to decidual transformation of stromal cells. The authors need to be more transparent about the phase/state of the samples they use and carefully phrase their conclusions accordingly.

3. The human endometrial stromal cell line (hESC), is poorly described. Where did this come from? Has it been validated, or used in other publications? There appears to be big differences in the level of senescence, so I would question the usefulness of using and comparing senescence across both?

4. The results section needs more rationale for performing the experiments. At present it is hard to follow and to understand why the authors performed certain experiments. This negatively impacts the overall clarity and interpretability of the scientific manuscript. The authors need to provide a clear and well-defined rationale for conducting each experiment, so readers understand the context of the results presented. Whereas this was present in places, it was absent elsewhere or needed to be further developed.

5. The authors used 8-br-cAMP throughout culture experiments to induce decidualization. There is some evidence that this non-physiological hit of cAMP is the main cause of cellular senescence, or at the very least enhances it. This needs to be addressed somewhere in the manuscript. Or alternatively, repeat some of the basic senescence-biomarkers using a physiological relevant induction of cAMP - such as PGE2.

6. For the uNK-cell co-culture experiment: There needs to be a clear distinction between clearing senescent-cells and inhibiting the induction of senescence. For the co-culture experiment it could be either, and the authors need to word this carefully as they frequently interchange between the 2 throughout the manuscript. Fig 4A is a bit confusing as it is unclear what days we are looking at (are we comparing just day 6?).

Minor concerns:

1. Please perform densitometry and show all average data for western blots.

It is also common to show the entire uncropped western blot in the supplementary section, however this is journal specific and up to the editor to decide if necessary.

2. uNK Cell isolation in methods: MACS not MASC

3. Cells were grown without any supplements. Can the authors confirm if this is true? Cultured cells often require L-glutamine

supplementation which is absent in DMEM/F12. Were any antibiotics used? Could the authors also confirm whether the FBS was stripped of hormones? Endometrial stromal cells are obviously sensitive to hormones

4. There was several typos and structuring issues throughout, I suggest a thorough proof-read.

Referee #2:

Gaining a better understanding of the mechanisms regulating stromal cell decidualization, homeostasis and pregnancy establishment could significantly impact women's reproductive health. The study by Shi et al. utilizes genetic, molecular, and pharmacological approaches to investigate the role of SLC3A2 and dNK in maintaining senescent homeostasis during decidualization. The manuscript, which holds potential interest for reproductive biologists and clinicians addressing fertility issues, could be improved by addressing the following points:

1. The manuscript would benefit from careful editing to improve grammatical clarity and rectify instances of missing words.
2. The utilization of previously published SC-RNA seq data necessitates a more detailed exposition of the data selection and analysis process in this study, including specifics on cell numbers, replicates, and data processing techniques.
3. Although cAMP is commonly used to induce decidualization in vitro, validating increased senescence with PGE2, a more physiological inducer of decidualization would be beneficial. This can be done using publicly available sc-RNA seq data (Stadtmauer and Wagner 2022).
4. The authors do not provide a citation for the aging model. Additionally, the authors should consider the model's potential effects on oocyte quality and embryo development. Employing embryo transfer could effectively address these concerns.
5. The manuscript needs a comprehensive expansion of its methodologies section, including but not limited to:
 - Preparation protocols for mouse decidual tissue in flow cytometry analysis.
 - Detailed RNA-seq preprocessing and downstream analysis.
 - Elaboration on sc-RNA Seq analysis, as previously mentioned.
 - Quantification methodologies for Western blot analysis.
6. Metformin has been shown to decrease decidual senescence, as shown by Sun et al. (2018), and the observed increase in decidual senescence in mouse models with compromised fertility due to DNA repair deficiencies, should be acknowledged and discussed within the manuscript.
7. It is difficult to decipher the labeling in the immunofluorescent images presented. Many of the labels cover the scale bars.

Referee #3:

The research article (EMBOJ-2023-116531) entitled "LIGHT+ NK cell prevents miscarriage mediated by excessive senescence of decidual stromal cell" by Shi et al. submitted to EMBO Journal aims to evaluate the role of LIGHT+ NK cells on decidual senescence. The authors indicate that increased decidual senescence can induce spontaneous abortion. It is well-known that endometrial stromal cells are the most abundant cell type in decidua during early pregnancy. The stromal cells dominate the communications with other cell types including immune cells, endothelial cells. Thus, aberrant decidualization is crucial for pregnancy maintenance. The research reveals the function of decidual cells and their communications with other cell type in normal and spontaneous aborted decidua pregnancy that provides insightful information for the pathology of early pregnancy loss.

Therefore, the authors' idea is original as well as the authors performed several experiments to prove their hypothesis. However, some experiments need to be repeated or reanalyzed because of inconsistency. The manuscript is very poorly written, needs to improve and revise many sentences by correcting many grammatical errors. The HSCORE data should be clearly given.

Additionally, please see my comments in below:

- In introduction, the authors should clearly clarify that decidualization is a progesterone dependent differentiation of endometrial stromal cells to decidual stromal cells (DSCs) in decidua that has a heterogenous mix of DSCs, leukocytes including NK cells and trophoblasts.
- Defective decidualization is not only linked to pregnancy loss and preeclampsia but also fetal growth restriction. Please add FGR with suitable references.
- The sentence "Of note, the increased autophagy and adhesion ability of stromal cells during decidualization leads to extensive infiltration and enrichment of NK cells, especially the TNF superfamily member 14 (TNFSF14, also named LIGHT)+ NK cells (Lu et al., 2021)" should be given later, because it is not connected to the sentences.
- Also, provide a kind of sentence "the TNF superfamily member, LIGHT, known as TNFSF14 and a T-cell costimulatory

molecule, is a critical ligand for the activation of NK cells."

- The authors should provide a clear hypothesis in the introduction.
- Brighton et al previously displayed similar results: 1) decidualization induces senescence in ESCs and 2) decidual NL cells mediate immune clearance of senescent decidual cells. However, this study was not discussed and cited. The authors should discuss similarity and differences between these two studies.
- In Results, the authors found that "DSC displayed the increased level of senescence (Figure 1G)" However, no information, these cells are decidualized or not, how long treated with cAMP+MPA. Also, the sentence should be corrected as " DSC displayed increased SABG staining (Figure 1G) versus ESC indicating elevated senescence in DCS."
- In Figure 2A, 1) how to determine decidual cells, did the authors perform immunostaining for decidual cell marker such as Vimentin in serial sections obtained from normal or spontaneous abortion material; 2) did the author perform HSCORE analysis to quantified immunostained slide?; 3) if quantified immunostaining, how did they score since cells display immunopositivity for both cytoplasm and nucleus; 4) how many slides immunostained in normal and SA; and 5) negative control slide should be added.
- In Figure 2B, the bar representing densitometric reading should be included.
- In Figure 2A, they found increased SLC3A2 and oxidative stress marker during decidualization. Please provide which control group is used (untreated??), and cells are how long decidualized.
- In Figure 2B, the undecidualized group as a control should be included to display increase expression of p21, p16 and p53 mRNA levels since the increase in these genes seems to be 0.25-fold only based on Figure 2A. Also, please use gene names in graphs.
- Immunoblotting in Suppl Fig 2D and E is unclear, which one is total which one is phosphorylated p38?
- The authors mentioned that "there were the higher expression of SLC3A2 and intake of BCAA by DSCs from SA patients, compared to DSCs from women with normal pregnancy (Figure 3F,G).? However, Figure 3F indicate higher SLC3A2 immunostaining in NP-decidua versus SA decidua. HSCORE should be added.
- The authors mentioned that "we observed high rates of embryo loss, low weights of embryos and placenta in high-leucine-fed pregnant mice (Figure 3K,L)." but no data provided in Figure 3K and L indicating lower fetal/placental weight. Fetal/placental weight and resorption rate should be given as a figure or table with mean{plus minus}SD or SEM.
- Fetal/placental weight and resorption rate should be given as a figure or table with mean{plus minus}SD or SEM for Figure 4F. The authors should be explained why they did not use WT female mated with WT male as a control or why Hvem-/- mated with Hvem-/- male did not use.
- If coculturing of DSC with dNK cells reduced decidual senescence, the authors should provide reduced number of NK cells in decidua from SA samples.
- Please provide full strain name of mice as C57/BL.
- Please provide a reference indicating use of D-galactose (120mg/kg/d for 8 weeks for aging process.
- How many mice were used in study?
- Which day of gestation i.p NK1.1 neutralizing antibody was injected; this is one time or injected during tissue collection at E13.5.
- There is no information generation of Hvem knockout mice. Should be provided or added references.
- What is the purity of endometrial stromal cells (ESC) or decidual stromal cells (DSC) or dNK?
- Which methods are used to determine the purity of these cells?
- Which passage numbers are used to perform the experiment?
- Related references should be added to cell isolation?
- dNK cells isolation should be expended.
- The author mentions that "DICs were in 40/60% interface" but no information of DICs?
- Why hESC cell line is used if the authors have primary cultured ESC. It is unclear which experiments were repeated in which cell type?
- For decidualization, the authors used 0.5 mM cAMP, this is so high concentration. Please explain? General concentration is 5×10^{-5} M during decidualization, please see DeMayo F' s studies.
- Why the author did not use estradiol in combination with MPA and cAMP during in vitro decidualization.
- Gene and protein names should be presented as NCBI Gene nomenclature in the revised manuscript.
- -All abbreviations in the manuscript should be used in first cited sentence and be consistent through manuscript.

Dear editors,

Thank you very much for your letter enclosing the comments for our manuscript entitled "LIGHT⁺ NK cell prevents spontaneous abortion by restricting leucine-mediated decidual stromal cell senescence" (EMBOJ-2023-116531). We resubmit a revised manuscript in which the revisions have been highlighted. The following are the responses to comments and suggestions about the manuscript.

We wish to take this opportunity to express our gratitude for your reconsideration of our paper for publication in your journal, *EMBO Journal*.

To Reviewer #1:

Referee #1:

The authors have used secondary and primary cells, murine and human ex vivo models to study cellular senescence in the endometrium. They have identified one mechanisms of cellular senescence by studying the BCAA transporter SLC3A2, and recognized that uNK cells regulate senescence through the LIGHT-HVEM interactions. A lot of work characterizing senescence was performed comparing control patients to those with a history of spontaneous abortions, implicating these mechanistic pathways in reproductive outcomes.

Whilst the work is performed to a high standard and is topical in the field of endometrial senescence, I feel the authors need to address the major points raised below in order to substantiate their conclusions.

Major concerns:

1. The laboratory work investigating the cellular and ex vivo studies is of high quality, and the murine model adds to the study. However, a large portion of the work relies on comparing human control endometrial biopsies with those from patients with unexplained spontaneous abortions (SA). The methods and source of endometrial biopsies is unclear.

More clarity is needed in the patient and sampling collection of the methods.

• How has the SA group been diagnosed? Do they suffer from recurrent SA? If so, how many? Do you have information on how long ago the spontaneous abortions occurred?

Response: Thanks for your kind suggestion. According to the 2017 ESHRE guidelines, the patients with recurrent miscarriage (RSA) in this study experienced two or more consecutive natural miscarriages before the 24th week of pregnancy. All cases were histologically verified based on established criteria. Pregnancies were confirmed through ultrasound and blood tests. Women experiencing spontaneous miscarriages attributed to endocrine, anatomical, and genetic issues, or to any infection, were excluded from the study. We have made supplements in the section of results and methods.

• How were the biopsies for cultures/uNKs timed to the secretory phase? Urinary LH? Patient-

reported cycling? Senescence and uNK phenotype changes dramatically over the secretory phase so authors need to ensure fair comparisons. How can the authors be sure that changes were not temporal?

Response: Thank you. The NK cells used in our research were decidual NK cells, which were isolated from decidual tissue during early normal pregnancy. To maintain phenotypic consistency of the decidual NK cells, we specifically targeted the collection period between 7 to 9 weeks of gestation. The detailed details have already been supplemented in the methods section.

2. The authors are using pregnant endometrium to compare control and SA, but at no point in the manuscript is this acknowledged. Instead, conclusions are extrapolated to decidual transformation of stromal cells. The authors need to be more transparent about the phase/state of the samples they use and carefully phrase their conclusions accordingly.

Response: Thank you. The transformation of the endometrium into decidua is known as decidualization. We compared cellular senescence before and after embryo implantation using control endometrial tissue from non-pregnant women and decidual tissue from women with normal pregnancy. Additionally, we developed an *in vitro* decidualization model of endometrial stromal cells (ESCs) to investigate the senescence of ESCs during this process. Furthermore, we contrasted the senescence of decidual stromal cells (DSCs) in the first trimester decidua from women with normal pregnancies (NP) and those with recurrent spontaneous abortion (RSA) to assess senescence under healthy and pathological conditions. Supplementary explanations have been incorporated into the manuscript to elucidate these aspects.

3. The human endometrial stromal cell line (hESC), is poorly described. Where did this come from? Has it been validated, or used in other publications? There appears to be big differences in the level of senescence, so I would question the usefulness of using and comparing senescence across both?

Response: Thank you for the kind advice. Human endometrial stromal cells (hESCs) were sourced from the American Type Culture Collection (CRL-4003; ATCC, Manassas, VA, USA), which have been previously utilized and validated in our research as indicated by the references (PMID: 36482461; PMID: 36185612). Concurrently, this study employed primary isolated ESCs to induce *in vitro* decidualization and assess cell senescence-related molecules, as depicted in Figure 1F. These results were consistent with the data derived from hESCs. The corresponding methodological details have been appropriately expanded upon in the methods section of our manuscript.

4. The results section needs more rationale for performing the experiments. At present it is hard to follow and to understand why the authors performed certain experiments. This negatively impacts the overall clarity and interpretability of the scientific manuscript. The authors need to provide a clear and well-defined rationale for conducting each experiment, so readers understand the

context of the results presented. Whereas this was present in places, it was absent elsewhere or needed to be further developed.

Response: Thank you for your suggestions. We sincerely apologize for any confusion or reading challenges that may have arisen from our initial description. We understand the importance of providing a clear and well-defined rationale for each experiment to ensure the scientific integrity and interpretability of our work. To address your comments, we have revised the manuscript to include a more comprehensive explanation for the design and purpose of each experiment. Specifically, we have made the following changes (line 46- line 584) as follow.

1. In the "Introduction" section, we have expanded upon the scientific background and the gaps in the current literature that our study aims to address. And we have also suggested a hypothesis. This provides a foundation for the reader to understand the necessity of our experimental approach.
2. We have added corresponding purposes and specific methods under the "Methods" section. For example, we have added the purpose and specific methods of animal experiments, single-cell sequencing, etc
3. For each experiment described in the "Results" section, we have now included a brief explanation of the hypothesis being tested and the relevance of the findings to our overarching research goals.
4. In the "Discussion" section, we have provided explanations for the results in the article, as well as the conclusions inferred from the results.

5. The authors used 8-br-cAMP throughout culture experiments to induce decidualization. There is some evidence that this non-physiological hit of cAMP is the main cause of cellular senescence, or at the very least enhances it. This needs to be addressed somewhere in the manuscript. Or alternatively, repeat some of the basic senescence-biomarkers using a physiological relevant induction of cAMP - such as PGE2.

Response: Thank you. Numerous studies have utilized 8-br-cAMP to induce decidualization (PMID: 31965050, 30374127, 12216111). This compound has also been employed in investigations exploring stromal cell senescence (PMID: 29227245). Consider your suggestion, we used PGE₂ plus MPA to induce decidualization of hESCs, and we detected the cell senescence-related molecules CDKN2A, CDKN1A and TP53 using Western Blot analysis. The results showed that the expression of CDKN2A, CDKN1A, and TP53 in the cells also increased after the hESCs were treated with PGE₂ to induce decidualization, which was consistent with the cAMP treatment. The results have been supplemented in Figure S1A.

6. For the uNK-cell co-culture experiment: There needs to be a clear distinction between clearing senescent-cells and inhibiting the induction of senescence. For the co-culture experiment it could be either; and the authors need to word this carefully as they frequently interchange between the 2 throughout the manuscript. Fig 4A is a bit confusing as it is unclear what days we are looking at (are we comparing just day 6?).

Response: Thank you. This study underscores the role of NK cells in curbing the senescence

of decidual stromal cells, thus preserving a balanced and stable state between stromal cell aging and decidual transformation at the maternal-fetal interface. Taking your feedback into consideration, we have revised the manuscript comprehensively. Moreover, existing literature indicates that NK cells can eliminate senescent cells through the secretion of interferon, a point we have now included in the discussion section.

We induced decidualization in hESCs and introduced decidual NK cells for a 48-hour co-culture on the fourth day, followed by the collection of stromal cells for downstream experiments two days afterward (which corresponds to the sixth day) (FigS3A). We have added pertinent supplementary explanations in both the methods and results sections to clarify this process.

Minor concerns:

1. Please perform densitometry and show all average data for western blots.

It is also common to show the entire uncropped western blot in the supplementary section, however this is journal specific and up to the editor to decide if necessary.

Response: Thank you. We have added the statistical chart of average protein expression. And we also provided raw data as requested by the journal.

2. uNK Cell isolation in methods: MACS not MASC

Response: Thank you. We have revised it.

3. Cells were grown without any supplements. Can the authors confirm if this is true? Cultured cells often require L-glutamine supplementation which is absent in DMEM/F12. Were any antibiotics used? Could the authors also confirm whether the FBS was stripped of hormones? Endometrial stromal cells are obviously sensitive to hormones

Response: Thank you. We used the DMEM/F12 - Dulbecco's Modified Eagle Medium bought from Hyclone. It contains 2.5 mM L-glutamine. And we also added 1% Penicillin/Streptomycin/Neomycin Solution. The cell growth needs FBS. To exclude the influence of hormones, HyClone Charcoal/Dextran Treated FBS (US origin, Heat Inactivated) was used in this study. We have supplemented it in the methods.

4. There was several typos and structuring issues throughout, I suggest a thorough proof-read.

Response: Thank you. We have carefully checked and revised the manuscript based on the reviewers' suggestions, and got the manuscript edited by a professional editing service (<https://webshop.elsevier.com/>) and Prof. Tao Zhang from Chinese University of Hong Kong.

Copy of certificate for reviewers removed

To Reviewer #2:

Referee #2:

Gaining a better understanding of the mechanisms regulating stromal cell decidualization, homeostasis and pregnancy establishment could significantly impact women's reproductive health. The study by Shi et al. utilizes genetic, molecular, and pharmacological approaches to investigate the role of SLC3A2 and dNK in maintaining senescent homeostasis during decidualization. The manuscript, which holds potential interest for reproductive biologists and clinicians addressing fertility issues, could be improved by addressing the following points:

1. The manuscript would benefit from careful editing to improve grammatical clarity and rectify instances of missing words.

Response: Thank you. We have carefully checked and revised the manuscript based on the reviewers' suggestions, and got the manuscript edited by a professional editing service (<https://webshop.elsevier.com/>) and Prof. Tao Zhang from Chinese University of Hong Kong.

2. The utilization of previously published SC-RNA seq data necessitates a more detailed exposition of the data selection and analysis process in this study, including specifics on cell numbers, replicates, and data processing techniques.

Response: Thank you. The specific information of the establishment, quality control, and analysis of single-cell sequencing related data has been reported in our previous research (PMID: 36482461). This study retrieved publicly available data from our research group and conducted differential gene analysis. We have added citations and explanations in the text.

3. Although cAMP is commonly used to induce decidualization in vitro, validating increased senescence with PGE2, a more physiological inducer of decidualization would be beneficial. This can be done using publicly available sc-RNA seq data (Stadtmauer and Wagner 2022).

Response: Thank you. Taking your suggestion into account, we employed a combination of

PGE2 and MPA to induce decidualization in hESCs. We then utilized Western Blot analysis to assess the expression of cell senescence-related proteins CDKN2A, CDKN1A, and TP53. Our findings indicate that the levels of these proteins were elevated following PGE₂-induced decidualization, which was consistent with the cAMP treatment. This additional data is now presented in Figure S1A. Furthermore, we have referenced Stadtmayer and Wagner's research within the revised manuscript.

4. The authors do not provide a citation for the aging model. Additionally, the authors should consider the model's potential effects on oocyte quality and embryo development. Employing embryo transfer could effectively address these concerns.

Response: Thank you. We have added citation in the methods (PMID: 26771610; PMID: 33565551). Consider your suggestion, we performed embryo transfer in the mice. In short, the fertilized eggs of normal C57 pregnant mice were transplanted into mice pre-injected with D-galactose and control. The specific experimental methods have been supplemented in the methods section (line 353-378). The results were shown in figure 2D-2G, and pregnant mice with excessive senescence of uterus were prone to embryo loss, along with the low number of blastocyst implantation. Additionally, the modifications have been made in the results and discussion sections.

5. The manuscript needs a comprehensive expansion of its methodologies section, including mbut not limited to:

- *Preparation protocols for mouse decidual tissue in flow cytometry analysis.*
- *Detailed RNA-seq preprocessing and downstream analysis.*
- *Elaboration on sc-RNA Seq analysis, as previously mentioned.*
- *Quantification methodologies for Western blot analysis.*

Response: Thank you for your suggestions. We sincerely apologize for any difficulties in reading caused by our inadequate description. We understand the importance of providing a clear and well-defined rationale for each experiment to ensure the scientific integrity and interpretability of our work. To address your comments, we have revised the manuscript to include a more comprehensive explanation for the design and purpose of each experiment. Specifically, we have made the following changes (line 330- line 598) as follow.

We have made corresponding supplements in the section of methods.

1. The specific protocols of mouse decidual tissue in flow cytometry were supplemented in the section of flow cytometry assays.
2. Detailed RNA-seq preprocessing and downstream analysis were supplemented in the section of RNA-seq.
3. Detailed sc-RNA Seq analysis were supplemented in the section of single-cell sequencing.
4. Quantification methodologies for Western blot analysis were supplemented in the section of western blotting.
5. The specific steps of animal experiments have also been supplemented in the section of mice.
6. The DSC and NK cell isolation and cell treatment were also supplemented in the section of cell isolation and cell treatment.

6. *Metformin has been shown to decrease decidual senescence, as shown by Sun et al. (2018), and the observed increase in decidual senescence in mouse models with compromised fertility due to DNA repair deficiencies, should be acknowledged and discussed within the manuscript.*

Response: Thank you. We have made supplements in the discussion as follow.

“In addition, previous study suggested that metformin could attenuate preterm birth in mice through alleviating premature aging of the decidua (Sun et al., 2018). And a lot of researches considered that the mechanisms of metformin attenuates cell senescence including reducing ROS production in mitochondrial complex I, regulating insulin and IGF-1 signaling and mTOR signaling, improving DNA repair, and inhibition of NF- κ B signal transduction induced by pro-inflammatory cytokines (Chen et al., 2022, Sunjaya et al., 2021). DNA repair disorders are closely related to reproductive senescence and decreased fertility (Sonowal et al., 2023). Therefore, metformin should prevent spontaneous abortion by multiple pathways, including the decrease of cell senescence and increase of FBP accumulation in decidua, improve DNA repair ability, regulating mTOR signaling process, etc, that needs to conduct more in-depth research in the future”.

7. *It is difficult to decipher the labeling in the immunofluorescent images presented. Many of the labels cover the scale bars.*

Response: Thank you. We have revised it according to your kind suggestion.

To Reviewer #3:

Referee #3:

The research article (EMBOJ-2023-116531) entitled "LIGHT+ NK cell prevents miscarriage mediated by excessive senescence of decidual stromal cell" by Shi et al. submitted to EMBO Journal aims to evaluate the role of LIGHT+ NK cells on decidual senescence. The authors indicate that increased decidual senescence can induce spontaneous abortion. It is well-known that endometrial stromal cells are the most abundant cell type in decidua during early pregnancy. The stromal cells dominate the communications with other cell types including immune cells, endothelial cells. Thus, aberrant decidualization is crucial for pregnancy maintenance. The research reveals the function of decidual cells and their communications with other cell type in normal and spontaneous aborted decidua pregnancy that provides insightful information for the pathology of early pregnancy loss.

Therefore, the authors' idea is original as well as the authors performed several experiments to prove their hypothesis. However, some experiments need to be repeated or reanalyzed because of in consistency. The manuscript is very poorly written, needs to improve and revise many sentences by correcting many grammatical errors. The HSCORE data should be clearly given. Additionally, please see my comments in below:

Response: We would like to express our sincere gratitude for the time and effort you have dedicated to reviewing our manuscript. Your insights and constructive criticism are highly valued and will significantly help us improve the quality of our work. We have carefully

checked and revised the manuscript based on the reviewers' suggestions, and got the manuscript edited by a professional editing service (<https://webshop.elsevier.com/>) and Prof. Tao Zhang from Chinese University of Hong Kong. Additionally, we have modified and supplemented the related sections according to your suggestion.

- *In introduction, the authors should clearly clarify that decidualization is a progesterone dependent differentiation of endometrial stromal cells to decidual stromal cells (DSCs) in decidua that has a heterogenous mix of DSCs, leukocytes including NK cells and trophoblasts.*

Response: Thank you. We have made relevant supplements in the section of Introduction (line 74-77).

- *Defective decidualization is not only linked to pregnancy loss and preeclampsia but also fetal growth restriction. Please add FGR with suitable references.*

Response: Thanks for your helpful advice. We have added it in the revised manuscript (line 50-54).

- *The sentence "Of note, the increased autophagy and adhesion ability of stromal cells during decidualization leads to extensive infiltration and enrichment of NK cells, especially the TNF superfamily member 14 (TNFSF14, also named LIGHT)⁺ NK cells (Lu et al., 2021)" should be given later, because it is not connected to the sentences.*

Response: Thank you. We have revised it in the revised manuscript (line 82-85).

- *Also, provide a kind of sentence "the TNF superfamily member, LIGHT, known as TNFSF14 and a T-cell costimulatory molecule, is a critical ligand for the activation of NK cells."*

Response: Thank you. We have made supplements in the manuscript (line 85-87).

- *The authors should provide a clear hypothesis in the introduction.*

Response: Thank you. We have made supplements in the text (line 92-94).

- *Brighton et al previously displayed similar results: 1) decidualization induces senescence in ESCs and 2) decidual NK cells mediate immune clearance of senescent decidual cells. However, this study was not discussed and cited. The authors should discuss similarity and differences between these two studies.*

Response: Thank you. We have added relevant supplements in the section of discussion (line 227-230 and 271-274).

- *In Results, the authors found that "DSC displayed the increased level of senescence (Figure 1G)"*

However, no information, these cells are decidualized or not, how long treated with cAMP+MPA. Also, the sentence should be corrected as " DSC displayed increased SABG staining (Figure 1G) versus ESC indicating elevated senescence in DCS."

Response: Thanks for your kind suggestion. In the figure 1G, we isolated endometrial stromal cells (ESCs) from endometrial tissue during secretion phase, and decidual stromal cell (DSCs) from the decidua in the first trimester. In this experiment, there was no treatment with cAMP+MPA. And then compared the SA β G activity between ESCs and DSCs. We have made modification in the text.

- In Figure 2A, 1) how to determine decidual cells, did the authors perform immunostaining for decidual cell marker such as Vimentin in serial sections obtained from normal or spontaneous abortion material; 2) did the author perform HSCORE analysis to quantified immunostained slide?; 3) if quantified immunostaining, how did they score since cells display immunopositivity for both cytoplasm and nucleus; 4) how many slides immunostained in normal and SA; and 5) negative control slide should be added.

Response: Thank you. We initially stained the decidual tissue using an anti-human vimentin antibody to ascertain that the large, rounded cells present within the decidual tissue are indeed decidual stromal cells (PMID: 20075392). And the immunofluorescence results of vimentin also showed that large and round cells in decidual tissue were decidual stromal cells (the result was as follows). Then we conducted immunohistochemical experiments on the tissue and determined DSC based on cell morphology. The results of immunohistochemistry showed that the expression of CDKN2A, CDKN1A, and TP53 on normal decidual stromal cells is lower than that on decidual stromal cells in the RSA group. Gene cards and our immunohistochemistry results both showed that CDKN2A, CDKN1A and TP53 existed in the cytoplasm and nucleus. We also invited pathology experts to help us determine the localization of stromal cells. In order to further compare the expression of CDKN2A, CDKN1A, and TP53 in DSC between normal and RSA decidua tissues in a more intuitive way, we extracted proteins from DSC of two groups for WB experiments, and used image J to calculate the Relative expression of target proteins (Fig 2B). The slides of immunostained in normal pregnancy and spontaneous abortion is 6 respectively, which is showed in the figure legend. And we have added negative control slide.

Figure for reviewers removed

- In Figure 2B, the bar representing densitometric reading should be included.

Response: Thank you. We have added it.

- In Figure 2A, they found increased *SLC3A2* and oxidative stress marker during decidualization. Please provide which control group is used (untreated??), and cells are how long decidualized.

Response: Thank you. The bubble plot of Figure 3A is based on single-cell sequencing of secretory endometrial stromal cells (ESCs) and early pregnancy decidual stromal cells (DSCs). We have provided additional explanations in figure legend.

- In Figure 2B, the undecidualized group as a control should be included to display increase expression of *p21*, *p16* and *p53* mRNA levels since the increase in these genes seems to be 0.25-fold only based on Figure 2A. Also, please use gene names in graphs.

Response: Thank you. Maybe you meant Fig 3B? In the Fig3B, we used different culture medium during the decidualization of hESCs to explore the influence of BCAA in the stromal cell senescence, and the culture medium (DMEM/F12) as the control. And we have revised the gene names in graphs.

- Immunoblotting in Suppl Fig 2D and E is unclear, which one is total which one is phosphorylated p38?

Response: Thank you. The upper band is P38, the lower band is phosphorylated p38 (PP38). We have supplemented the detailed information in the revised manuscript.

- The authors mentioned that "there were the higher expression of *SLC3A2* and intake of BCAA by DSCs from SA patients, compared to DSCs from women with normal pregnancy (Figure 3F,G).? However, Figure 3F indicate higher *SLC3A2* immunostaining in NP-decidua versus SA decidua. HSCORE should be added.

Response: Thanks for your help suggestion. We have replaced the images with more representative ones, and added the HSCORE.

- The authors mentioned that "we observed high rates of embryo loss, low weights of embryos and placenta in high-leucine-fed pregnant mice (Figure 3K,L)." but no data provided in Figure 3K and L indicating lower fetal/placental weight. Fetal/placental weight and resorption rate should be given as a figure or table with mean{plus minus}SD or SEM.

Response: Thank you. The right side of the Fig 3L (that is Fig.4D in revised manuscript) shows the statistics of embryo and placenta quality (Data were presented as mean \pm SEM).

- Fetal/placental weight and resorption rate should be given as a figure or table with mean{plus minus}SD or SEM for Figure 4F. The authors should be explained why they did not use WT female mated with WT male as a control or why *Hvem*^{-/-} mated with *Hvem*^{-/-} male did not use.

Response: Thank you. Fig 4G showed the statistical results of fetal embryo absorption rate, embryo and placental weight (Data were presented as mean ± SEM). According to the previous study (PMID: 33030400), in order to ensure the same fetal genotype and exclude the influence of fetal factors on the results, we used *Hvem*^{-/-} pregnant mice obtained by combining *Hvem*^{-/-} female mice with WT male mice as the experimental group, and WT pregnant mice obtained by combining WT female mice with *Hvem*^{-/-} male mice as the control group.

- If coculturing of DSC with dNK cells reduced decidual senescence, the authors should provide reduced number of NK cells in decidua from SA samples.

Response: Thank you. In our study, we suggested that decidual NK cells inhibited stromal cell senescence. And we also detected the expression of LIGHT in the decidual NK cells and HVEM in the DSCs in the decidua of normal pregnancy and recurrent spontaneous abortion. And we found that the percent of all NK cells and the LIGHT⁺NK is no difference in the above two groups. In RSA group, the expression of HVEM in DSC was decreased. Therefore, in the co-culture system of NK cells and stromal cells, cell number of stromal cells and NK cells is consistent in each group. Among these, we silenced the expression of TNFRSF14 (HVEM) in DSCs, and further co-cultured with decidual NK cells, and then analyzed the senescence level of DSCs.

- Please provide full strain name of mice as C57/BL.

Response: Thank you. We have revised it.

- Please provide a reference indicating use of D-galactose (120mg/kg/d for 8 weeks for aging process).

Response: Thank you. We have supplemented it in the method.

- How many mice were used in study?

Response: Thank you. The number of mice of every experiment were pointed in the ever figure legend. The total number of female mice represent in the figure are 56.

- Which day of gestation i.p NK1.1 neutralizing antibody was injected; this is one time or injected during tissue collection at E13.5.

Response: Thank you. NK1.1 neutralizing antibody was injected began the 0th day of pregnancy, and then inject every two days. That is to say, injections are administered on the 0th, 2nd, 4th, 6th, 8th, 10th, and 12th day of pregnancy. We have supplemented it in the method.

- *There is no information generation of Hvem knockout mice. Should be provided or added references.*

Response: Thank you. The *Hvem* knockout mice were generated by knocking out the exon 2 region of the *Hvem* gene. The *Hvem* knockout mice (Cat. NO. NM-KO-190170) were purchased from Shanghai Model Organisms Center, Inc. We have provided corresponding supplementary explanations in the section of methods.

- *What is the purity of endometrial stromal cells (ESC) or decidual stromal cells (DSC) or dNK?*

Response: Thank you. First, we isolated the DSC from decidual by Percoll (cells between concentration 20% and 40% were DSCs, between 40% and 60% were DICs), and then DSCs were cultured in the culture flask for one night to allow the DSCs adhere and remove other cells. And DICs were also cultured in the culture flask for one night in order to remove other cells. And then supernatant culture medium was collected to isolate dNK cells. The ESCs isolated from endometrium, and cells were cultured in the culture flask for one night to allow the ESCs adhere and remove other cells. The purity of DSCs and ESCs were more than 95%, and the purity of dNK cells were more than 90%. We have supplemented in the methods.

- *Which methods are used to determine the purity of these cells?*

Response: Thank you. Flow cytometry was used to verify the cell purity.

- *Which passage numbers are used to perform the experiment?*

Response: Thank you. In this study, the passage number for all hESC lines was limited to no more than six generations, we have supplemented the information in revised manuscript.

- *Related references should be added to cell isolation?*

Response: Thank you. We have added it.

- *dNK cells isolation should be expanded.*

Response: Thank you. We have added it.

- *The author mentions that "DICs were in 40/60% interface" but no information of DICs?*

Response: Thank you. We isolated DIC from decidual tissue and then isolated NK cells from DICs. We have provided additional explanations in the NK cell section.

- Why hESC cell line is used if the authors have primary cultured ESC. It is unclear which experiments were repeated in which cell type?

Response: Thank you. We found that a lot of researches used hESC to explore decidualization (PMID: 34591094; PMID: 26003431; PMID: 14726435). And our previous studies have used and validated the hESCs (PMID: 36482461; PMID: 36185612). In addition, we also detected the expression of senescence related molecules in primary ESCs during decidualization (Fig. 1F). The results were consistent with the hESC cell line, thus we used hESC to perform further researches.

- For decidualization, the authors used 0.5 mM cAMP, this is so high concentration. Please explain? General concentration is $5 \times 10^{-5} M$ during decidualization, please see DeMayo F' s studies.

Response: Thank you for your kind suggestion. Brighton PJ. et al used cAMP (0.5mM) plus MPA (1 μ M) to induce decidualization, in order to explore the stromal cell senescence (PMID: 29227245). And another study also constructed the ESC decidualization model (0.5mM cAMP plus 1 μ M MPA) for single-cell sequencing to study the cell differentiation mechanism (PMID: 31965050). In addition, a large number of researchers used 0.5 mM cAMP to construct the decidualization model (PMID: 30374127; PMID: 12216111).

- Why the author did not use estradiol in combination with MPA and cAMP during in vitro decidualization.

Response: Thank you. The combined induction of decidualization by estrogen and progesterone needs 6-8 days, while the combined treatment of cAMP+MPA has a higher efficiency in inducing decidualization. Currently, a large number of studies have used cAMP+MPA combined treatment to study decidualization (PMID: 30374127; PMID: 29227245). In addition, we also used PGE₂+MPA for auxiliary verification, which is added in the section of figure supplement.

- Gene and protein names should be presented as NCBI Gene nomenclature in the revised manuscript.

Response: Thank you. We have revised it.

- All abbreviations in the manuscript should be used in first cited sentence and be consistent through manuscript.

Response: Thank you for your kind suggestion. We have revised it.

Dear Prof. Li,

Congratulations on a great revision! Overall, the referees have been positive. However, referee 3 has raised a few minor concerns that we ask you to (non-experimentally) address in a revised version. When you submit your revised version, please also take care of the following editorial items and add this also to your point-by-point response:

1. Please provide an author checklist.
2. Please remove the figures from the main manuscript file.
3. Please use an institutional email address for author Zhao.
4. Please move the Data Availability section to the end of the methods and add URLs that go directly to the datasets for GSE183837, GSE194219, SAMN39254954, SAMN39254955, SAMN39254956, and SAMN39254957. Please add the database name for RNA-sequencing data.
5. 2023YFC2705403 is in eJP but not listed in the manuscript, please add this funding information.
6. Please remove the author contribution section from the main manuscript.
7. The file with the supplementary tables and figures should be renamed Appendix and uploaded in PDF format and with the yellow highlights in the text removed. It needs a table of contents (with page numbers). The tables should be renamed "Appendix Table S1" and S2, and the figures should be "Appendix Figure S1" etc.
8. We include a synopsis of the paper (see <http://emboj.embopress.org/>). Please provide me with a general summary statement and 3-5 bullet points that capture the key findings of the paper.
9. We also need a summary figure for the synopsis. The size should be 550 wide by 200-440 high (pixels). You can also use something from the figures if that is easier.
10. The list of supplementary materials should be removed from the manuscript.
11. Please note that the supplementary figure 5a-c does not contain any quantification graph, kindly rectify the statistics related information in the figure legend appropriately.
12. Please define the annotated p values **/* as well as provide the exact p-values for the same in the legend of supplementary figure 1a; 3c; as appropriate.
13. Please note that the exact p values are not provided in the legends of figures 1e-g; 2b-c, f-g; 3b-g; 4a-b, d, h-i; 5b-d, f-g; 6c-d, f-i; 7a-b, d; 8c, e-g, supplementary figure 2a; 4c; 6b; 7a-b.
14. Please indicate the statistical test used for data analysis in the legends of figures 1e-g; 2b-c, f-g; 3b-g; 4a-b, d-f, h-i; 5b-d, f-g; 6c-d, f-i; 7a-b, d; 8c, e-g, supplementary figure 1a; 2a; 3c; 4c; 6b; 7a-b.
15. Although 'n' is provided, please describe the nature of entity for 'n' in the legends of figures 1e-g; 2b-c, f-g; 3b-g; 4e-f, h-i; 5c-d, f-g; 6c-d, f-i; 8a, f-g, supplementary figure 1a; 2a; 3c; 4c.
16. Please note that the error bars are not defined in the legends of supplementary figure 1a; 3c.
17. It appears that the western provided in Figure 2B for CDKN2A does not match the source data you provided but rather looks like a duplicate of the TP53 western. Please provide the corrected Figure 2.

Thank you for the opportunity to consider your work for publication, I look forward to your revision.

Yours sincerely,

Kelly M Anderson, PhD
Editor, The EMBO Journal
k.anderson@embojournal.org

Referee #1:

The authors have addressed the majority of my concerns in my original review so I can recommend the manuscript for publication.

Referee #2:

The authors have adequately addressed my comments and this version is much improved.

Referee #3:

The research article (EMBOJ-2023-116531R) entitled "TNFS14+ NK cells prevent spontaneous abortion by restricting leucine-mediated decidual stromal cell senescence" by Shi et al. resubmitted to EMBO Journal aims to evaluate the role of LIGHT+ NK cells on decidual senescence. Authors responded to my points raised in the original submission. However, there are still minor points outstanding.

L107, what is the difference between human ESCs and primary ESC, please explain?? Or are you mentioning ESCs and DSCs??

L111-114, as requested by the Reviewers, the authors tested the effect of PGE2 on senescence marker in hESCs (Fig. S1A), but cells were treated with MPA + PGE2. It is difficult to understand that enhanced CDKN2A, CDKN1A and TP53 levels are associated with MPA or PGE2? Also, please use full name of PGE2. What is the difference between Figure 1D and E?? if there is, please explain detailly in figure legends.

L122-128, the authors should add their findings based on Fig. 2F such as reduced number of implantation rate with increased absorption rate

Referee #3:

The research article (EMBOJ-2023-116531R) entitled "TNFS14+ NK cells prevent spontaneous abortion by restricting leucine-mediated decidual stromal cell senescence" by Shi et al. resubmitted to EMBO Journal aims to evaluate the role of LIGHT+ NK cells on decidual senescence. Authors responded to my points raised in the original submission. However, there are still minor points outstanding.

1. L107, what is the difference between human ESCs and primary ESC, please explain?? Or are you mentioning ESCs and DSCs??

Response: Thank you. At this point in the text, we are indicating that both a human endometrial stromal cell line and primary endometrial stromal cells (ESCs) were utilized to develop our decidualization model. The term 'human ESCs' refers to the cell line, and this clarification has been incorporated into the revised manuscript.

2. L111-114, as requested by the Reviewers, the authors tested the effect of PGE2 on senescence marker in hESCs (Fig. S1A), but cells were treated with MPA + PGE2. It is difficult to understand that enhanced CDKN2A, CDKN1A and TP53 levels are associated with MPA or PGE2? Also, please use full name of PGE2. What is the difference between Figure 1D and E?? if there is, please explain detailly in figure legends.

Response: Thank you. In this study, we tried to stromal cell senescence levels during decidualization, we constructed in vitro decidualization models of human ESC (hESC) cell line and primary ESCs. As illustrated in the figure below, the standard in vitro induction of decidualization is achieved through treatment with medroxyprogesterone acetate (MPA) combined with 8-bromo-cAMP, or MPA in conjunction with PGE2.

Table for reviewers removed

This is the previous round of reviewer comment (*Although cAMP is commonly used to induce decidualization in vitro, validating increased senescence with PGE2, a more physiological inducer of decidualization would be beneficial. This can be done using publicly available sc-RNA seq data (Stadtmauer and Wagner 2022)*). As shown in Figure 1 of Stadtmauer and Wagner's research, a combination of MPA and PGE2, rather than MPA alone, was effective in inducing decidualization. So MPA plus PGE2 was also used to induce decidualization in this research (Figure 4 of Stadtmauer and Wagner's research).

Our study did not specifically focus on the effects of MPA, PGE2, or cAMP, we concur that these agents serve as methods for mimicking decidualization. Our conclusion is that stromal cell senescence is enriched during the decidualization process, as supported by the data from Stadtmauer and Wagner (2022) and our in vitro models. According to your kind suggestion, we have revised the figure 8H, removed E2+P4, and changed to decidualization.

Figures for reviewers removed

Figure IE showed the another cell senescence indicator SA β G activity during the decidualization. We have supplemented it in the figure legend.

3. L122-128, the authors should add their findings based on Fig. 2F such as reduced number of implantation rate with increased absorption rate

Response: Thanks for your helpful advice, we have added it.

Dear Ming-Qing,

Congratulations on an excellent manuscript, I am pleased to inform you that your manuscript has been accepted for publication in The EMBO Journal. Thank you for your comprehensive response to the referee concerns and for providing detailed source data. It has been a pleasure to work with you to get this to the acceptance stage.

I will begin the final checks on your manuscript before submitting to the publisher next week. Once at the publisher, it will take about 3 weeks for your manuscript to be published online. As a reminder, the entire review process, including referee concerns and your point-by-point response, will be available to readers.

I will be in touch throughout the final editorial process until publication. In the meantime, I hope you find time to celebrate!

Warm wishes,
Kelly

Kelly M Anderson, PhD
Editor, The EMBO Journal
k.anderson@embojournal.org

>>> Please note that it is The EMBO Journal policy for the transcript of the editorial process (containing referee reports and your response letter) to be published as an online supplement to each paper. If you do NOT want this, you will need to inform the Editorial Office via email immediately. More information is available here: https://www.embojournal.org/transparent-process#Review_Process